# The speed of GTP hydrolysis determines GTP cap size and controls microtubule stability

Johanna Roostalu[1‡]*, Claire Thomas[1†], Nicholas Ian Cade[1†], Simone Kunzelmann[1], Ian A Taylor[1], Thomas Surrey[1,2,3]*

[1]The Francis Crick Institute, London, United Kingdom; [2]Centre for Genomic Regulation, Barcelona Institute of Science and Technology, Barcelona, Spain; [3]ICREA, Barcelona, Spain

**Abstract** Microtubules are cytoskeletal polymers whose function depends on their property to switch between states of growth and shrinkage. Growing microtubules are thought to be stabilized by a GTP cap at their ends. The nature of this cap, however, is still poorly understood. End Binding proteins (EBs) recruit a diverse range of regulators of microtubule function to growing microtubule ends. Whether the EB binding region is identical to the GTP cap is unclear. Using mutated human tubulin with blocked GTP hydrolysis, we demonstrate that EBs bind with high affinity to the GTP conformation of microtubules. Slowing-down GTP hydrolysis leads to extended GTP caps. We find that cap length determines microtubule stability and that the microtubule conformation changes gradually in the cap as GTP is hydrolyzed. These results demonstrate the critical importance of the kinetics of GTP hydrolysis for microtubule stability and establish that the GTP cap coincides with the EB-binding region.

*For correspondence:
johanna@roostalu.info (JR);
thomas.surrey@crg.eu (TS)

[†]These authors contributed equally to this work

Present address: [‡]The Wellcome Trust, London, United Knigdom

## Introduction

The dynamic nature of microtubules is critical for their function in cells. Microtubules are hollow tubes that polymerize by the addition of GTP-bound α/β-tubulin heterodimers to their ends. Tubulin incorporation induces GTP hydrolysis which destabilizes the microtubule wall. A delay in hydrolysis of unknown duration is thought to produce a 'GTP cap', a protective end structure formed of GTP-tubulins that is critical for microtubule stability (*Brouhard and Rice, 2018*; *Carlier, 1982*; *Erickson and O'Brien, 1992*; *Howard and Hyman, 2009*; *Mitchison and Kirschner, 1984*). The loss of this cap is thought to expose an unstable GDP lattice and trigger depolymerization, resulting in dynamic instability of microtubule growth (*Mitchison and Kirschner, 1984*). How the biochemistry of GTP hydrolysis translates into conformational changes in the growing microtubule end, and thereby determines the properties of the GTP cap and microtubule stability, is still not understood (*Brouhard and Rice, 2018*; *Kueh and Mitchison, 2009*).

End binding proteins of the EB family (EBs) bind autonomously to an extended region at growing microtubule ends where the GTP cap is expected to be (*Bieling et al., 2008*; *Bieling et al., 2007*; *Dixit et al., 2009*; *Komarova et al., 2009*; *Maurer et al., 2012*; *Roth et al., 2018*). The affinity of EBs for binding to the growing microtubule end region is about 10 fold higher than to the GDP part of the microtubule lattice at a distance from the end (*Maurer et al., 2011*; *Seetapun et al., 2012*). In cells, EBs control microtubule dynamics and function by locally recruiting numerous regulators of cytoskeleton function specifically to growing microtubule ends (*Akhmanova and Steinmetz, 2015*).

Whether EBs bind indeed to the GTP cap is still debated, mostly because they bind differently to microtubules grown in the presence of different non-hydrolysable GTP analogues. EBs bind to

GTPγS microtubules with similar affinity than to growing microtubule ends, but considerably less to GMPCPP microtubules (*Maurer et al., 2011*; *Roth et al., 2018*). This is probably a consequence of structural differences between microtubules grown with these two GTP analogues (*Alushin et al., 2014*; *Manka and Moores, 2018*; *Zhang et al., 2015*; *Zhang et al., 2018*). However, in contrast to GTPγS, GMPCPP strongly promotes microtubule nucleation and stabilizes microtubules, and is therefore usually considered to be the better GTP analogue for microtubules (*Alushin et al., 2014*; *Hyman et al., 1992*; *Manka and Moores, 2018*). These observations with GTP analogues and the detection of a very short EB-free zone at the very end of growing microtubules, just in front of the EB binding region, have raised the question whether EBs really bind the GTP cap or possibly instead a post-hydrolysis state in the growing microtubule end region (*Maurer et al., 2014*; *Zhang et al., 2015*). It is therefore unclear whether the EB binding region and the GTP cap are identical and which GTP analogue induces more faithfully the GTP state in microtubules. This limits our understanding of the nature of the GTP cap.

Here we studied recombinant human microtubules with altered GTP hydrolysis kinetics to address these unanswered questions. We found that microtubules with blocked GTP hydrolysis were hyper-stable, nucleated extremely efficiently and that EBs bound with high affinity along their entire length. Microtubules with impaired, but not blocked GTPase rate were also considerably more stable than wildtype microtubules and displayed higher nucleation efficiency. They had an elongated EB binding region at the growing microtubule ends, indicative of slowed-down GTP hydrolysis. Together, these results demonstrate that EBs do indeed bind with high affinity to the GTP conformation of the microtubule and consequently are GTP cap markers. Moreover, we observed an EB affinity gradient along the length of the GTP cap, indicative of a conformational and stability gradient within the cap as a result of GTP hydrolysis.

## Results

To examine the relationship between GTP hydrolysis and conformational changes in the GTP cap and to answer the question of the nucleotide preference of EBs, we produced recombinant human tubulin, which allowed us to alter the GTPase activity of microtubules. Human α/β-tubulin was expressed in insect cells and purified (and separated from endogenous insect cell tubulin) by affinity chromatography using an internal hexa-histidine-tag in α-tubulin (*Sirajuddin et al., 2014*; *Vemu et al., 2016*) and a protease-cleavable Strep-tag at the C-terminus of β-tubulin (*Figure 1—figure supplement 1*) We generated wildtype α/β-tubulin and a tubulin variant with blocked GTP hydrolysis by mutating the evolutionarily conserved catalytic glutamate 254 of α-tubulin (*Nogales et al., 1998*) to alanine (E254A) (*Figure 1a–b*, *Figure 1—figure supplement 1*). This mutation is lethal in budding yeast, as cell division fails due to loss of microtubule dynamics (*Anders and Botstein, 2001*).

We measured the GTP content of microtubules polymerized from purified recombinant human wildtype tubulin and tubulin purified from porcine brain as a control. We observed the expected 1:1 ratio of GTP:GDP (*Figure 1c–d*), since GTP is hydrolyzed only at its inter-tubulin dimer, but not at the intra-dimer binding site (*Kobayashi, 1975*). In contrast, E254A microtubules bound almost exclusively GTP (*Figure 1c–d*), demonstrating that GTP hydrolysis is indeed blocked in this mutant.

To visualize the growth dynamics of label-free recombinant microtubules elongating from surface-attached stable microtubule 'seeds' (*Figure 1—figure supplement 2a*), we used interferometric scattering (iSCAT) microscopy (*Ortega Arroyo et al., 2016*). In the presence of GTP, human wild-type microtubules grew dynamically (*Figure 1e–f*, *Video 1*) similarly to porcine brain microtubules (*Figure 1—figure supplement 2b–c*), displaying occasional transitions to depolymerization, called catastrophes, as expected (*Vemu et al., 2017*; *Vemu et al., 2016*).

Wildtype tubulin hardly nucleated any microtubules spontaneously (*Figure 1g*, *Videos 1* and *2*), whereas E254A tubulin nucleated microtubules very efficiently even at low concentrations, quickly reaching high microtubule densities (*Figure 1h*, *Video 3*). Moreover, these mutant microtubules were hyper-stable for hours, never exhibiting catastrophes. These properties agree with the anticipated behavior of hydrolysis-deficient microtubules, and are similar to the behavior of microtubules polymerized in the presence of the non-hydrolysable GTP analogue GMPCPP (*Hyman et al., 1992*).

Next, we tested how EBs bind to human wildtype and to hydrolysis-deficient E254A microtubules. Simultaneous total internal reflection fluorescence (TIRF) microscopy of mGFP-

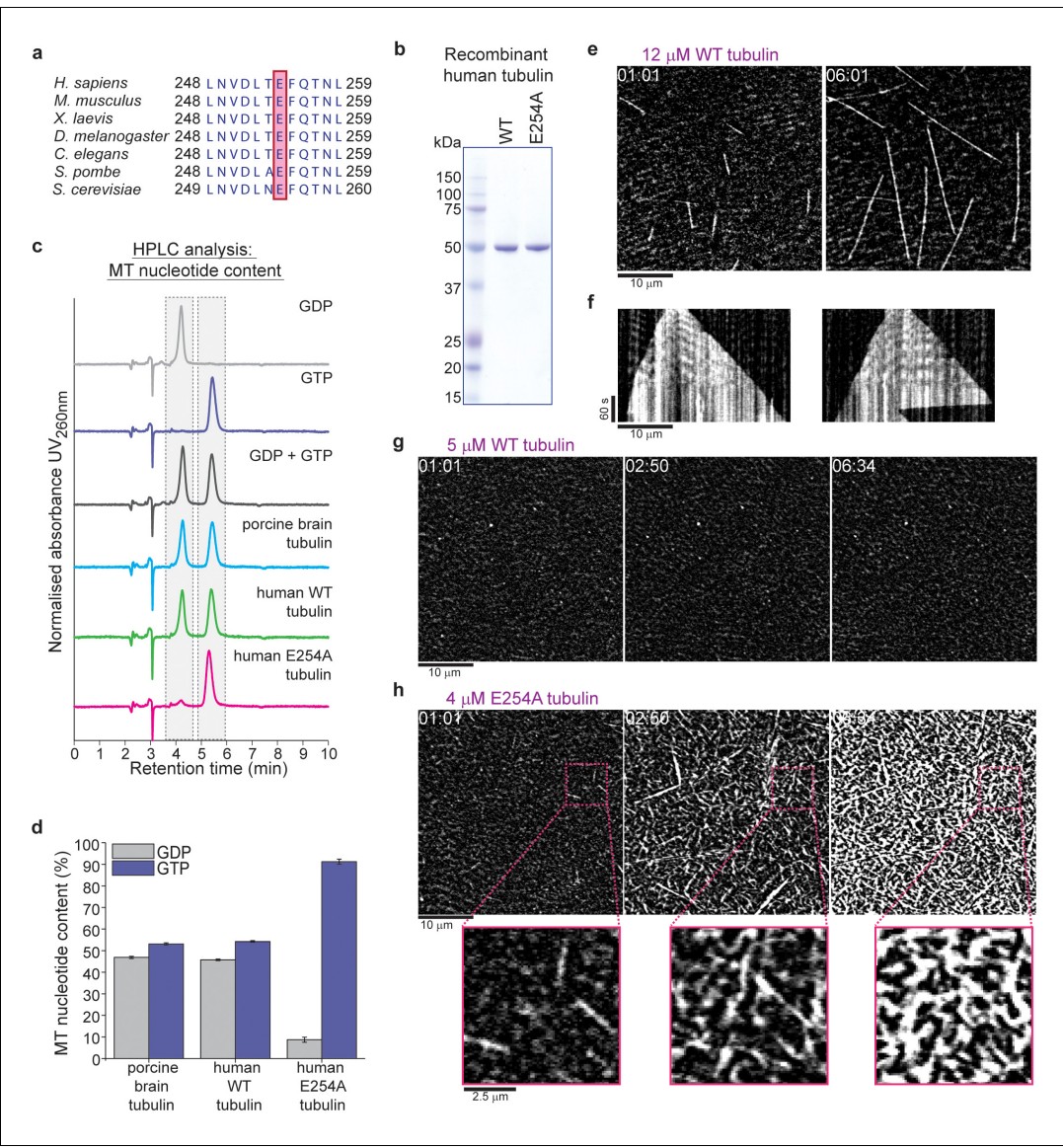

**Figure 1.** Recombinant human tubulin lacking GTPase activity strongly nucleates microtubules. (a) The evolutionarily conserved catalytic glutamic acid of α-tubulin (E, pink). (b) Coomassie Blue-stained SDS gel of purified wildtype (WT) and E254A mutant recombinant human tubulin. (c) HPLC chromatograms of pure GDP and GTP compared to nucleotides extracted from microtubules polymerized from porcine brain tubulin or wildtype and E254A mutant human tubulin. (d) Quantification of the nucleotide content of different microtubules. Bar graphs depict the means of 3 independent experiments, error bars represent standard deviation (SD). (e) iSCAT microscopy images of unlabeled wildtype human tubulin (12 μM) growing from immobilized GMPCPP-stabilized biotinylated seeds (GMPCPP-seeds). (f) Kymographs showing wildtype microtubule growth, conditions as in (e). (g, h) iSCAT microscopy images showing (g) lack of microtubule nucleation at 5 μM wildtype tubulin and (h) strong nucleation at 4 μM E254A tubulin at a surface with an immobilized rigor kinesin (Kin1[rigor]). Bottom: magnified E254A microtubule views. Scale bars as indicated, time is always min:sec.

The online version of this article includes the following figure supplement(s) for figure 1:

**Figure supplement 1.** Expression and purification of recombinant human α/β tubulin.
**Figure supplement 2.** iSCAT microscopy of unlabeled porcine brain microtubules.

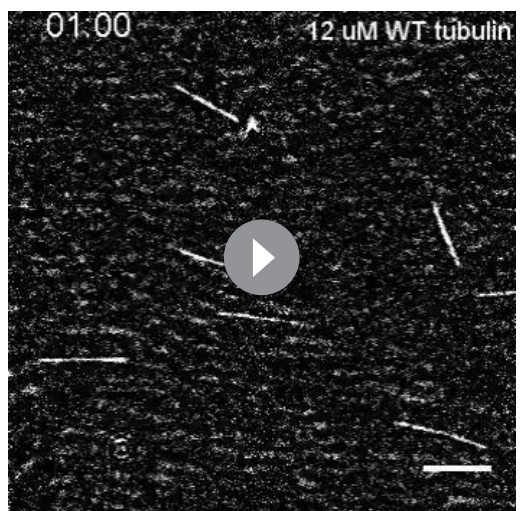

**Video 1.** Pure recombinant human wildtype tubulin polymerizes into dynamic microtubules. iSCAT microscopy time-lapse movie of unlabeled microtubules growing in the presence of 12 μM human wildtype tubulin from immobilized stable microtubule seeds at 30°C. Scale bar is 5 μm. Time stamp is in min:s.

https://elifesciences.org/articles/51992#video1

EB3 (*Figure 2—figure supplement 1a*) and iSCAT imaging of unlabeled microtubules revealed that EB3 bound to the growing plus and minus ends of wildtype microtubules in the expected comet-like manner (*Figure 2a–c*, *Video 4*; *Bieling et al., 2008*; *Bieling et al., 2007*; *Komarova et al., 2009*). The apparent dissociation constant for binding to growing microtubule ends was ~40 nM (*Figure 2d*). EB3 had no effect on the wildtype microtubule growth speed (*Figure 2e*).

In clear contrast, EB3 decorated the entire lattice of GTPase-deficient E254A mutant microtubules (*Figure 2f–h*; *Figure 2—figure supplement 1b*), reminiscent of its binding to microtubules grown in the presence of the non-hydrolysable GTP analogue GTPγS (*Maurer et al., 2011*). However, EB3 failed to bind human wildtype microtubules polymerized in the presence of GMPCPP (*Figure 2—figure supplement 1c*), as observed previously with mammalian brain microtubules (*Maurer et al., 2011*; *Roth et al., 2018*). This negative effect of GMPCPP on the EB3 binding affinity was also observed when E254A microtubules were grown in the presence of this GTP analogue (*Figure 2—figure supplement 1d*), demonstrating that it is indeed the bound nucleotide and not the single point mutation itself that controls the microtubule conformation and hence the affinity of EB3 binding. With an apparent dissociation constant of ~8 nM (*Figure 2i*), EB3 binding to E254A microtubules was stronger than to growing wildtype ends (*Figure 2d*). Unbinding of mGFP-EB3 from E254A microtubules was very slow (*Figure 2j–k*,

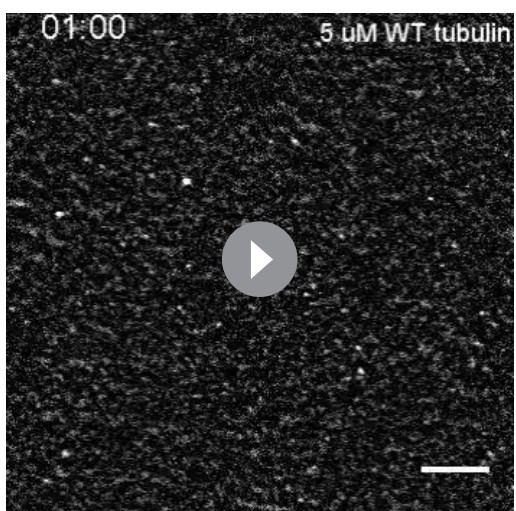

**Video 2.** Human wildtype tubulin does not nucleate microtubules at low tubulin concentration. iSCAT microscopy time-lapse movie showing lack of microtubule nucleation at 5 μM human wildtype tubulin on a surface with an immobilized rigor kinesin (Kin1[rigor]) at 30°C. Scale bar is 5 μm. Time stamp is in min:s.

https://elifesciences.org/articles/51992#video2

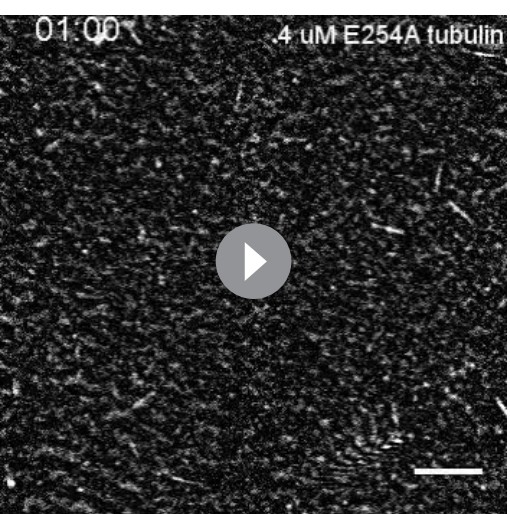

**Video 3.** Human E254A tubulin mutant lacking GTPase activity strongly nucleates microtubules. iSCAT microscopy time-lapse movie showing strong microtubule nucleation at 4 μM human E254A tubulin on a surface with an immobilized rigor kinesin (Kin1[rigor]) at 30°C. Scale bar is 5 μm. Time stamp is in min:s.

https://elifesciences.org/articles/51992#video3

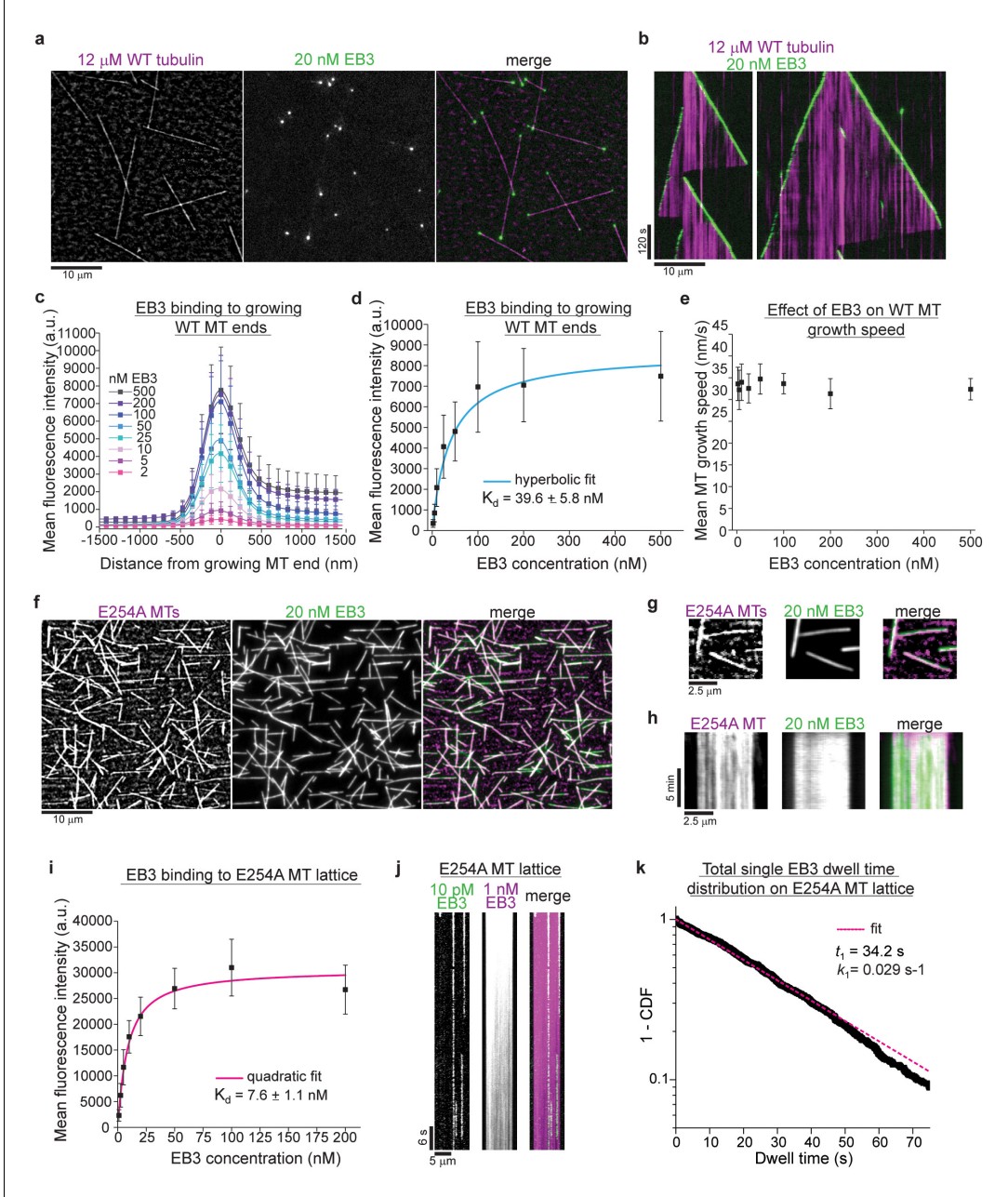

**Figure 2.** GTPase deficient microtubules are stable and bind EB3 with high affinity. (a) iSCAT/TIRF microscopy image of unlabeled wildtype microtubules (WT, magenta) growing from GMPCPP-seeds at 12 µM wildtype human tubulin in the presence of 20 nM human mGFP-EB3 (green). (b) Kymographs showing mGFP-EB3 (green) tracking the ends of growing wildtype microtubules, conditions as in (a). (c) Mean mGFP-EB3 intensity profiles at the ends of human wildtype microtubules (WT MT) growing at 12 µM wildtype tubulin in the presence of varying mGFP-EB3 concentrations (as indicated). Number of microtubules (and frames) averaged for each mGFP-EB3 concentration: 2 nM – 64 (7360), 5 nM – 87 (9222), 10 nM – 93 (8742), 25 nM – 89 (8900), 50 nM – 56 (5600), 100 nM – 58 (4756), 200 nM – 56 (4088), 500 nM – 46 (4140). Error bars are SD. (d) Quantification of mGFP-EB3 binding affinity to microtubule ends growing at 12 µM wildtype tubulin (WT MT). Black symbols depict the averaged maximal mGFP-EB3 intensities at growing plus ends at each condition, error bars represent SD. Experimental data as in (c). A hyperbolic fit (cyan) was used to calculate the $K_d$. (e) mGFP-EB3 does not affect significantly the growth speed of human wildtype microtubules (WT MT). Experimental data as in (c). (f) iSCAT/TIRF microscopy images of 20 nM mGFP-EB3 (green) binding to unlabeled E254A mutant microtubules (magenta) attached to a Kin1$^{rigor}$ surface. (g) Magnified images and (h) kymographs of stable unlabeled E254A microtubules (magenta) decorated with 20 nM mGFP-EB3 (green), conditions as in (d). (i) Quantification of mGFP-EB3 binding affinity to E254A mutant microtubules (E254A MTs). Black symbols depict the averaged mGFP intensities measured all along the microtubules at each condition, error bars represent SD. A quadratic fit (magenta) was used to calculate the $K_d$. Number of microtubules measured for each mGFP-EB3 concentration: 1 nM – 96, 2.5 nM – 126, 5 nM – 86, 10 nM – 69, 20 nM – 78, 50 nM – 61, 100 nM – 64, 200 nM - 75. (j) Kymograph of mGFP-EB3 (green) single molecule imaging at 10 pM on an individual E254A microtubule (MT) in the presence of 1 nM Alexa647-EB3 (magenta). (k)

*Figure 2 continued on next page*

*Figure 2 continued*

Dwell time analysis of mGFP-EB3 molecules on E254A microtubule (MT) lattice plotted as a survival function (1 - CDF, cumulative density function). Dashed line (magenta) is a mono-exponential fit to the data (black dots). Number of microtubules analyzed - 126, number of mGFP-EB3 binding events - 807. Scale bars as indicated.

The online version of this article includes the following figure supplement(s) for figure 2:

**Figure supplement 1.** Characterization of EB3 binding to wildtype and GTPase deficient human microtubules.

*Figure 2—figure supplement 1f–g*). Together these results show that EB3 displays strong affinity to microtubules locked in the GTP state, clearly establishing it as a *bona fide* GTP cap marker.

Interestingly, E254A microtubules simultaneously capture properties that different GTP analogues induce separately in wildtype microtubules: strong microtubule nucleation as in the presence of GMPCPP, and high EB binding affinity as in the presence of GTPγS (*Hyman et al., 1992*; *Maurer et al., 2011*). Another protein that discriminates between a GMPCPP microtubule and a growing microtubule end is a short fragment of the multifunctional anti-catastrophe factor TPX2 (*Wittmann et al., 2000*), previously called TPX2^micro (*Zhang et al., 2017*). Opposite to EBs, it displays higher affinity towards GMPCPP microtubules than towards growing microtubule ends (*Zhang et al., 2017*). We observed that this fragment also binds poorly to E254A microtubules compared to GMPCPP microtubules (*Figure 2—figure supplement 1e*), again suggesting that GTP-bound mutant microtubules are similar to growing microtubule ends. These results suggest that microtubules grown in GTP analogues display only partial aspects of the GTP conformation of microtubules, which explains previous contradictory interpretations based on results obtained with such analogues. In conclusion, a GTP-containing E254A mutant microtubule currently appears to be the best static mimic for the microtubule conformation in the only transiently existing GTP cap.

To investigate how GTP hydrolysis determines conformational transformations within the GTP cap, we generated a human tubulin mutant, in which glutamate 254 is substituted by aspartate (E254D) (*Figure 3a*). This chemically similar but shorter catalytic residue is expected to slow down GTP hydrolysis compared to wildtype microtubules. E254D microtubules grew steadily from surface-immobilized seeds (*Figure 3b*) and EB3 tracked their growing ends, displaying strikingly longer and brighter fluorescent 'comets' (*Figure 3c–d*, *Videos 5–7*), clearly indicative of slower GTP hydrolysis. E254D microtubules hardly ever displayed any catastrophes even under conditions where wildtype microtubules exhibited frequent depolymerization events (*Figure 3e–f*, *Videos 5–7*). E254D tubulin also nucleated microtubules more efficiently in solution than wildtype tubulin, but less than E254A tubulin (*Figure 3b*, *Figure 1h*) indicating an intermediate stability of E254D microtubules compared to wildtype and completely GTPase-deficient E254A microtubules. This suggests that GTP hydrolysis is indeed slowed down in the E254D mutant.

E254D microtubules grew about twofold faster than wildtype microtubules, either with or without EB3 (*Figure 3h*). A comparison of the average EB3 intensity profiles along the ends of wildtype and E254D microtubules growing at tubulin concentrations adjusted such that they grow with comparable speeds, shows that around ~5 times more EB3 binds to the ends of E254D microtubules (*Figure 3i*). This increase can to a large extent be explained by the longer EB3 binding region on the mutant microtubules (*Figure 3j*). Quantitative analysis of the intensity profiles allows to extract the characteristic comet lengths and, knowing the measured growth speeds, also the lifetime of the EB3 binding region (*Bieling et al., 2008*; *Bieling et al., 2007*; *Maurer et al., 2014*; *Maurer et al., 2012*), which was four-fold longer at the ends of E254D microtubules compared to wildtype microtubules (*Figure 3j*). These results demonstrate that slowed-down GTP hydrolysis extends the lifetime and hence the size of the GTP cap.

We occasionally observed 'split comets' at the growing ends of E254D microtubules that could later re-join (*Figure 3g*, magenta arrowheads) or detach and grow individually (*Figure 3g*, cyan arrowheads). This indicates that even partial microtubule end structures can be more stable when GTP hydrolysis is slow. This is reminiscent of recently observed split microtubule ends in the presence of the protofilament capping drug eribulin and the microtubule-stabilizing protein CLASP (*Aher et al., 2018*; *Doodhi et al., 2016*). The 'curved' appearance of these partial comets likely reflects global conformational differences between unfinished end structures such as observed for microtubule nucleation and fully formed tubes, pointing to a high degree of structural plasticity of

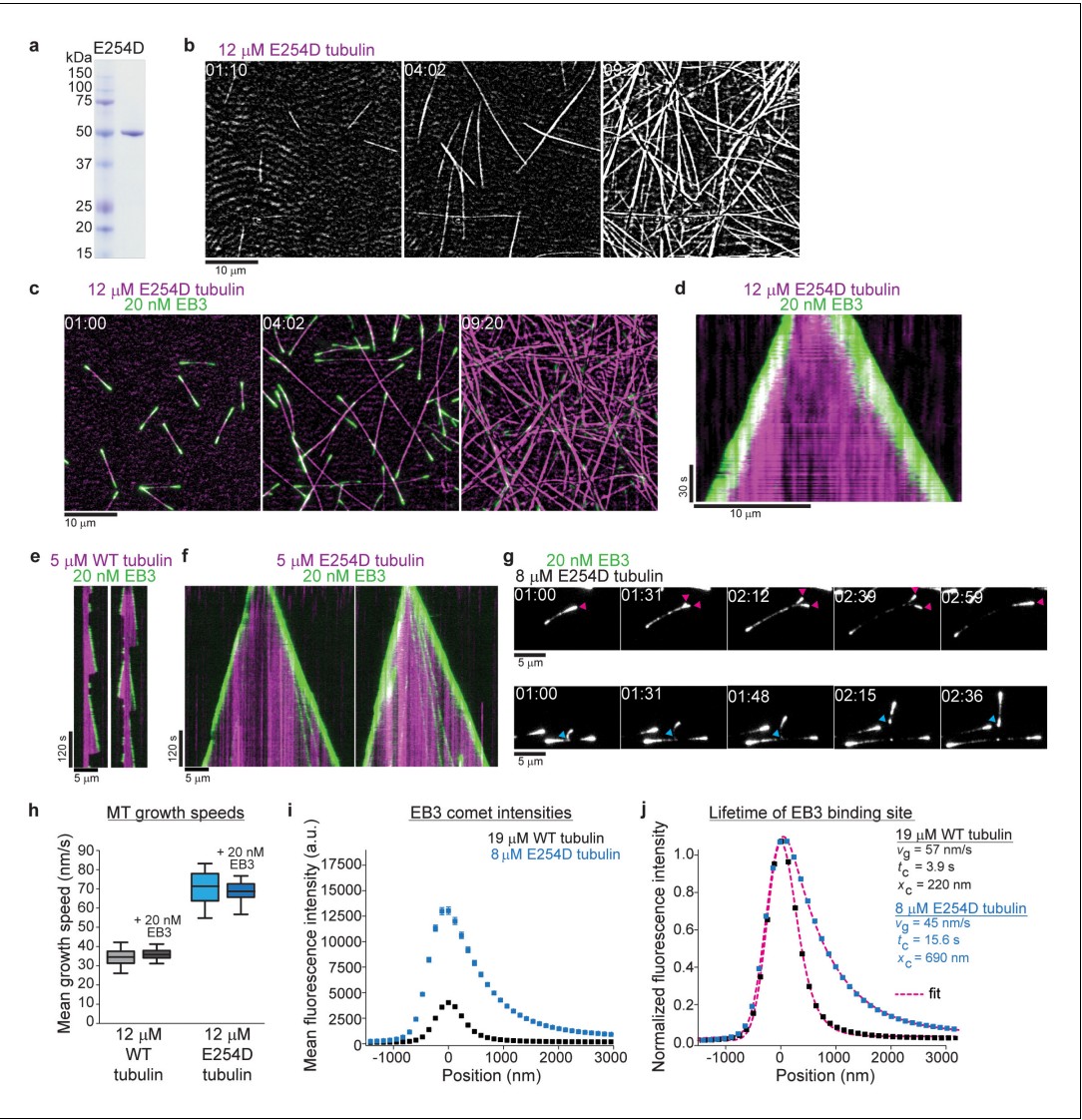

**Figure 3.** Slowing down GTP hydrolysis extends the GTP cap and stabilizes growing microtubules. (**a**) Coomassie Blue-stained SDS gel with purified E254D mutant recombinant human tubulin. (**b, c**) iSCAT/TIRF microscopy images of unlabeled E254D mutant microtubules growing from GMPCPP-seeds and also spontaneously nucleating at 12 μM E254D mutant human tubulin (**b**) in the absence and (**c**) presence of 20 nM human mGFP-EB3 (green). (**d**) Kymograph showing mGFP-EB3 (green) tracking the ends of persistently growing E254D microtubules, conditions as in (**c**). (**e, f**) iSCAT/TIRF microscopy kymographs depicting (**e**) wildtype (WT) and (**f**) E254D microtubules (magenta) growing at 5 μM tubulin in the presence of 20 nM mGFP-EB3 (green). E254D microtubules display very few catastrophes under conditions where wildtype human microtubules undergo frequent catastrophes. (**g**) TIRF microscopy images of protofilament bending, and re-association (arrowheads, magenta) or splitting (arrowheads, cyan) at unlabeled E254D microtubule ends growing at 8 μM tubulin visualized by mGFP-EB3. (**h**) Quantification of wildtype (WT) and E254D microtubule growth speeds at 12 μM respective tubulin concentrations in the presence and absence of mGFP-EB3. The boxes extend from 25th to 75th percentiles, the whiskers extend from 5th to 95th percentiles, and the mean value is plotted as a line in the middle of the box. Number of microtubules measured for each condition: 12 μM wildtype tubulin – 95, 12 μM wildtype tubulin and 20 nM mGFP-EB3 – 86, 12 μM E254D tubulin – 35, 12 μM E254D tubulin and 20 nM mGFP-EB3 - 83. (**i**) Mean mGFP-EB3 intensity profiles at growing wildtype (black) and E254D (blue) microtubule plus ends. Number of microtubules (and frames averaged) for each condition: wildtype – 68 (36960), E254D – 74 (41720). Error bars are SE. (**j**) Normalized comet profiles from (**h**) with dashed lines (magenta) representing exponentially modified Gaussian fits (see Materials and methods) yielding the comet length $x_c$ and, knowing the measured growth speed $v_g$, the life time $t_c = x_c/v_g$ of the EB binding sites. Scale bars as indicated, time is min:sec.

the GTP cap region (*Roostalu and Surrey, 2017*; *Voter and Erickson, 1984*; *Brouhard and Rice, 2018*; *Igaev and Grubmüller, 2018*).

How the GTP cap stabilizes growing microtubule ends is an open question. As GTP is hydrolyzed over time, a nucleotide state gradient may form at the growing microtubule end that in turn might translate into a conformational stability gradient. Such a gradient would be detectable as a gradient of EB affinities within the GTP cap. We therefore probed the microtubule conformation along the length of the GTP cap by measuring the binding strength of EB3 at different positions within the GTP cap using single molecule imaging and dwell time analysis (*Figure 4—figure supplement 1*). We first took advantage of the elongated GTP caps of growing E254D microtubules.

We imaged single mGFP-EB3 molecules binding in the elongated cap region of growing E254D microtubules, with their end regions labeled by excess Alexa647-EB3 (*Figure 4a*). We observed an overall complex (non-mono-exponential) distribution of mGFP-EB3 dwell times in the GTP cap, indicative of the presence of different conformational states with different EB3 binding affinities in the cap (*Figure 4b*). Next, we performed a spatially resolved dwell time analysis by generating several 'local dwell time distributions' for distinct distances from the growing microtubule end. These local distributions were rather mono-exponential, indicative of a more homogenous local conformational lattice state (*Figure 4c*). The local characteristic EB3 dwell times decreased along the length of the GTP cap (*Figure 4c–d*, *Figure 4—figure supplement 2a*). This indicates a decreasing EB binding affinity along the length of the GTP cap, suggesting a gradual conformational change within the cap. The long EB3 dwell times at the very tip of the GTP cap are indicative of a GTP-like state, given that EB3 binds strongly to GTPase deficient E254A microtubules (*Figure 2f–k*), which then transforms into the lower affinity GDP state concomitant with GTP hydrolysis. The conformational change along the length of the GTP cap most likely reflects a stability gradient whose characteristic length corresponds to the cap length (*Figure 4d*).

We observed a similar gradual conformational change in the shorter GTP caps of wildtype microtubules growing at a similar speed (*Figure 4—figure supplement 3*), when we recorded a considerably larger number of single molecule binding events compared to previous studies (*Bieling et al., 2008*; *Bieling et al., 2007*; *Dixit et al., 2009*; *Maurer et al., 2011*; *Maurer et al., 2014*) to enable spatially resolved dwell time analysis (*Figure 4e–h*). We detected also here a combination of conformational states within the whole GTP cap, as indicated by a complex EB3 dwell time distribution for all events observed in the entire cap region (*Figure 4f*), whereas rather homogenous conformational states were detected at distinct positions within the GTP cap, as indicated by more mono-exponential local dwell time distributions (*Figure 4g*). The mean dwell times, indicative of the EB3 binding affinity, decreased over a shorter distance in wildtype GTP caps (*Figure 4h*, *Figure 4—figure supplement 2c*) than in E254D caps (*Figure 4d*, *Figure 4—figure supplement 2a*), reflecting the shorter EB3 comet length at wildtype microtubule ends compared to longer EB3 comets at E254D microtubule ends with reduced GTPase activity (*Figure 3j*, *Figure 4—figure supplement 2b,d*). Taken together, these results suggest that a conformational gradient in the GTP cap at the growing microtubule end is a response to the GTP hydrolysis in the lattice.

## Discussion

Using recombinant mutated human tubulin we produced for the first time pure GTP-microtubules, eliminating the need for using non-hydrolysable GTP analogues. GTP-microtubules were hyper-stable, as predicted by the GTP cap hypothesis, and EBs bound strongly to these GTP microtubules. Microtubules with impaired GTP hydrolysis (E254D) were also more stable than wildtype microtubules and displayed longer EB comets, indicative of longer GTP caps. Our results are to date the most direct demonstration that the kinetics of GTP hydrolysis determine GTP cap length and that EBs bind to the GTP cap, explaining why their binding region represents the part of the microtubule that stabilizes it (*Duellberg et al., 2016*; *Maurer et al., 2012*).

The length of the GTP cap of recombinant human wildtype microtubules was in a similar range as previously measured EB 'comet' lengths with mammalian brain microtubules in vitro (*Bieling et al., 2008*; *Bieling et al., 2007*; *Dixit et al., 2009*; *Maurer et al., 2011*; *Rickman et al., 2017*; *Roth et al., 2018*; *Vemu et al., 2017*) and in cells (*Seetapun et al., 2012*). This confirms that recombinant human wildtype tubulin and mammalian brain tubulin hydrolyze GTP with similar speeds (*Vemu et al., 2017*). The hydrolysis rate appears to be set such that sufficiently long GTP caps are

created to stabilize growing microtubules during fluctuating growth (*Duellberg et al., 2016*; *Gardner et al., 2011*; *Rickman et al., 2017*; *Schek et al., 2007*).

Our results suggest that previous observations with GTP analogues require careful interpretation, because neither GMPCPP nor GTPγS generate microtubules with properties identical to microtubules that contain GTP. When considering microtubule stability and the affinity for EB binding, each analogue appears to produce some, but not all features of the microtubule lattice conformation in the GTP cap.

The recent observation of a short EB-free region at the very end of growing microtubules (*Maurer et al., 2014*) together with the observation of EBs not binding GMPCPP very well, had stimulated a discussion about the possibility of EBs binding a post-hydrolysis and not a GTP induced conformation of the microtubule lattice (*Maurer et al., 2014*; *Zhang et al., 2015*). Given the results presented here, this can now be considered unlikely. The observed short EB-free zone at the extreme microtubule end is probably rather a consequence of finite EB association kinetics (*Maurer et al., 2014*) and of some particular structural feature at the very end of growing microtubules. EB binding sites that are located between protofilaments (*Maurer et al., 2012*; *Zhang et al., 2015*), may not form there due to a flared growing microtubule end structure lacking lateral protofilaments interactions, as observed recently (*McIntosh et al., 2018*), or another GTP lattice conformation may exist at the very microtubule end.

For wildtype microtubules and microtubules with slowed-down GTP hydrolysis, we performed here for the first time spatially resolved dwell time measurements of single EB3 molecules at different positions of the GTP cap. Our aim was to gain insight into the conformational changes of the microtubule lattice within the cap that affect the conformation of the EB binding site, allowing it to monitor the lattice conformation. The binding strength of EB3 decreased gradually from the microtubule end towards the GDP lattice (*Figure 4d,h*), both for wildtype and E254D caps. Binding strength decayed mono-exponentially on a length scale similar to the comet lengths measured by 'comet' analysis (*Figure 3j*), suggesting that the density of EBs decreases along the length of the cap due to an affinity decrease. The associated time scale of this affinity decrease as the microtubule grows provides an estimate for the GTP hydrolysis rate which is ~0.2 s$^{-1}$ for wildtype microtubules and four times slower for E254D microtubules.

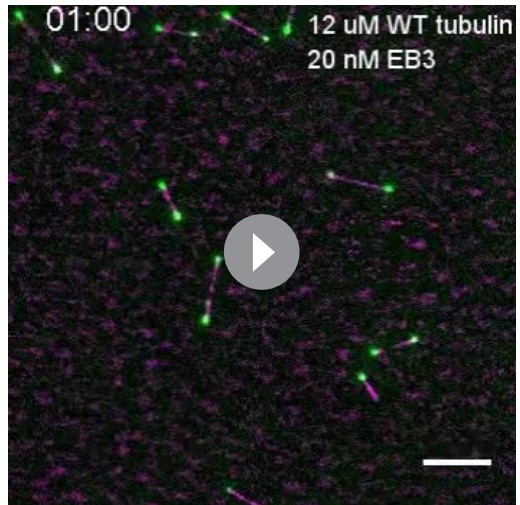

**Video 4.** EB3 binds to the growing ends of human wildtype microtubules. iSCAT/TIRF microscopy time-lapse movie of unlabeled wildtype microtubules (magenta) growing from immobilized stable microtubule seeds in the presence of 12 µM human wildtype tubulin and 20 nM human mGFP-EB3 (green) at 30℃. Scale bar is 5 µm. Time stamp is in min:s.
https://elifesciences.org/articles/51992#video4

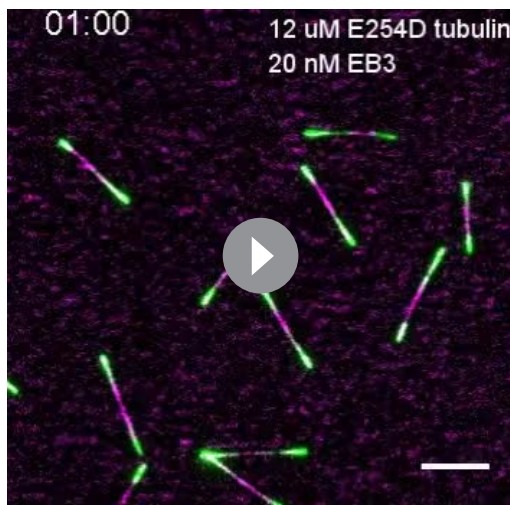

**Video 5.** Slowing down GTP hydrolysis extends the GTP cap and stabilizes growing microtubules. iSCAT/TIRF microscopy time-lapse movie of unlabeled E254D mutant microtubules (magenta) growing from immobilized microtubule seeds and also spontaneously nucleating in the presence of 12 µM human E254D mutant tubulin and 20 nM human mGFP-EB3 (green) at 30℃. Scale bar is 5 µm. Time stamp is in min:s.
https://elifesciences.org/articles/51992#video5

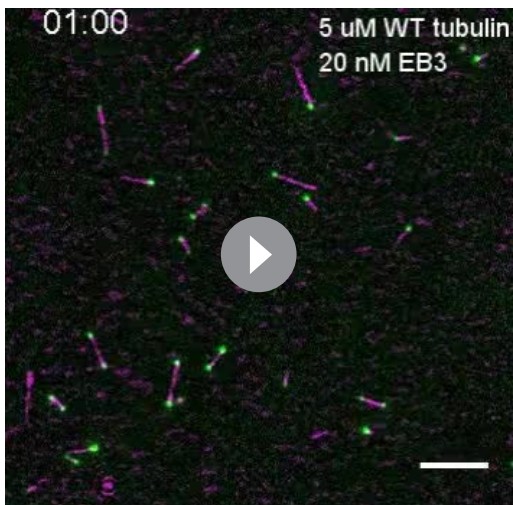

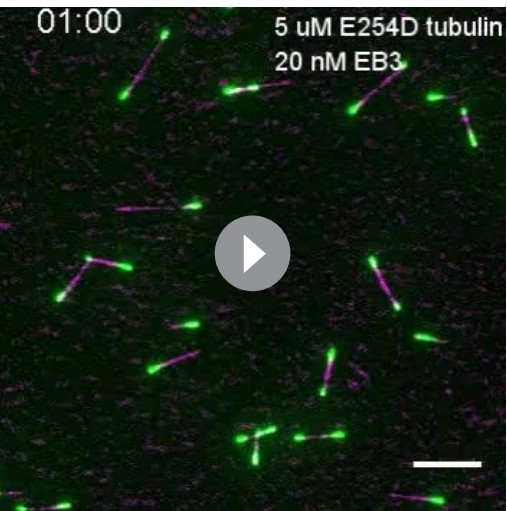

**Video 6.** Wildtype microtubules are very dynamic at low tubulin concentrations. iSCAT/TIRF microscopy time-lapse movie of human wildtype microtubules (magenta) undergoing frequent catastrophes when growing from immobilized microtubule seeds in the presence of 5 μM human wildtype tubulin and 20 nM human mGFP-EB3 (green) at 30℃. Scale bar is 5 μm. Time stamp is in min:s.

https://elifesciences.org/articles/51992#video6

**Video 7.** Microtubules with reduced GTPase activity are more stable than wildtype microtubules. iSCAT/TIRF microscopy time-lapse movie of human E254D microtubules (magenta) show persistent growth when growing from immobilized microtubule seeds in the presence of 5 μM E254D tubulin and 20 nM human mGFP-EB3 (green) at 30℃. Scale bar is 5 μm. Time stamp is in min:s.

https://elifesciences.org/articles/51992#video7

Using a mean squared displacement (MSD) analysis of single EB3 molecules bound to E254D caps we find that EB3 diffuses on the microtubule lattice with a diffusion coefficient of ~$10^{-3}$ μm²/s. Although this is several magnitudes slower than EB1 lattice diffusion reported previously under different conditions on microtubules grown in nucleotide analogues (*Lopez and Valentine, 2016*), it means that during a typical binding event EB3 samples the conformational state of several tubulins in a range of up to ~50 nm in our experiments. This defines the spatial resolution of this conformational reporter. We can not discriminate if all tubulins in the sampled area have exactly the same conformation or if EBs report on the average conformation of a lattice with individual tubulins being in different conformational states. Nevertheless, EBs clearly detect a conformational gradient within the GTP cap, as evidenced by the dependence of its mean dwell time on the distance from the growing microtubule end.

Interestingly, the ratio between the dwell times of EB3 bound to E254A microtubules (*Figure 2h–i*, *Figure 2—figure supplement 1f,g*) versus wildtype microtubule ends (*Figure 4f*) was considerably larger than the ratio between the corresponding equilibrium association constants (inverse equilibrium dissociation constants) (*Figure 2d and i*). This indicates that not only the unbinding rate, but also the binding rate of EB3 is reduced on E254A microtubules compared to wildtype microtubules, even if so to a lesser extent. This suggests that EB3 may induce and stabilize a conformational lattice transformation in E254A microtubules when it binds, akin to an 'induced fit' scenario. Such an effect on the microtubule lattice would agree with previous observations showing that EB binding to microtubules can change their protofilament twist (*von Loeffelholz et al., 2017*; *Zhang et al., 2015*; *Zhang et al., 2018*) and mildly accelerate cap maturation (*Maurer et al., 2011*). We can now conclude that the latter means 'accelerate GTP hydrolysis', in agreement with our results here as well as recent structural observations (*Zhang et al., 2015*).

In conclusion, our results demonstrate that the kinetics of GTP hydrolysis critically determine the properties of the GTP cap the size of which can be monitored by EBs. Because EBs serve as a binding platform for numerous microtubule binding proteins it ensures that microtubule dynamics in cells is regulated at multiple levels by targeting the most critical modulator of microtubule stability, the GTP cap, itself. Future challenges will be to elucidate the high-resolution structures of microtubules

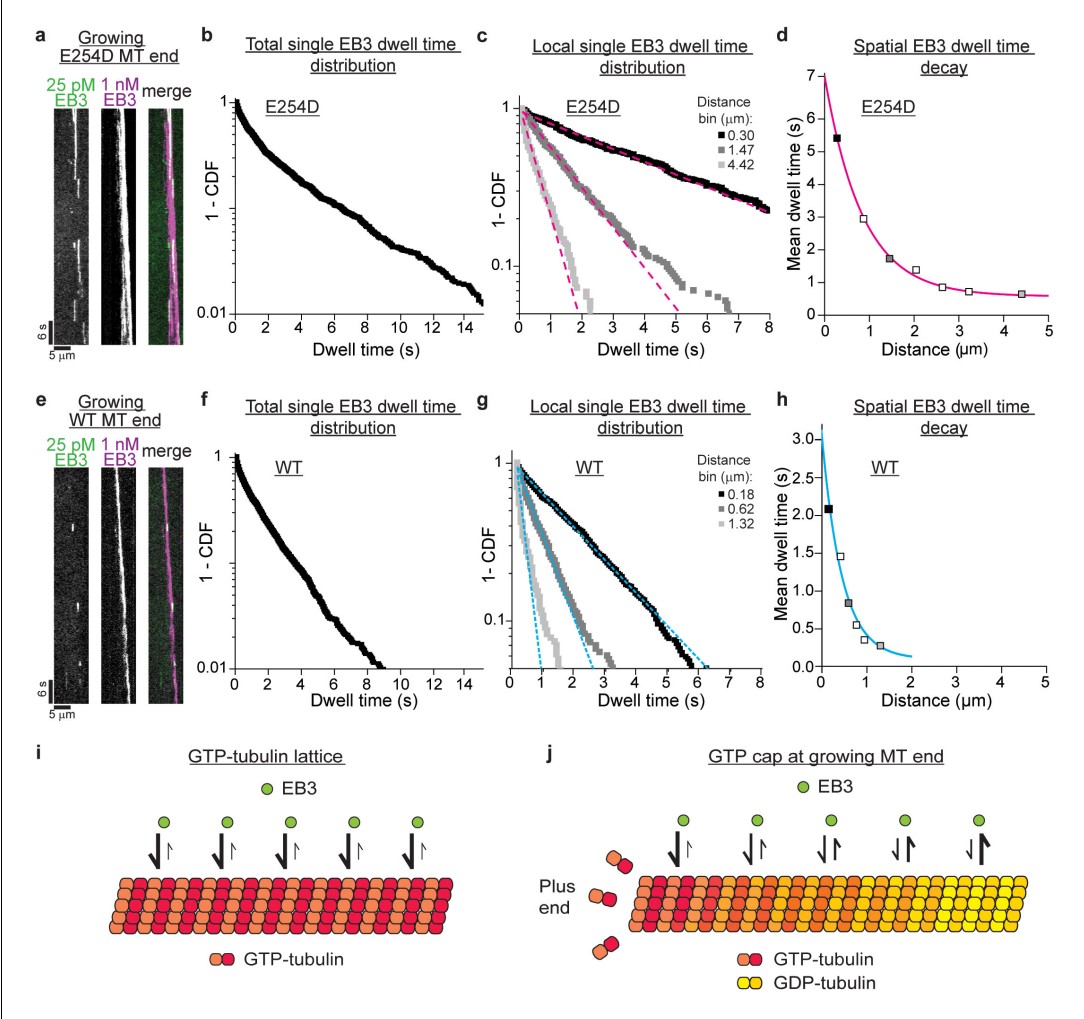

**Figure 4.** Conformational gradient in the GTP cap revealed by spatially resolved single EB3 dwell time distributions. (a) TIRF microscopy kymograph of single mGFP-EB3 molecules (green) at 25 pM in the plus end region of an E254D microtubule (E254D MT) growing at 10 μM E254D tubulin ($v_g$ = 58.5 nm/s) in the additional presence of 1 nM Alexa647-EB3 (magenta) (for end region visualization). (b) Dwell time distribution of single mGFP-EB3 molecules plotted as a survival function (1- CDF, cumulative density function). Number of microtubules analyzed - 150, number of mGFP-EB3 binding events - 1834. (c) Local dwell time distributions at distinct distances from the growing microtubule plus end. Dashed magenta lines are mono-exponential fits. The distance bins are 0–0.59, 1.18–1.77 and 3.84–5.02 μm; bin centers shown in the legend. Experimental data as in (b). (d) Local mean EB3 dwell times as a function of distance from the growing E254D microtubule end. Filled symbols correspond to the local mean dwell times calculated based on data in (c). The solid magenta line is a mono-exponential fit with a decay length of 850 nm. (e) TIRF microscopy kymograph of single mGFP-EB3 molecules (green) at 25 pM in the plus end region of a wildtype (WT) microtubule growing at 19 μM wildtype tubulin ($v_g$ = 59.8 nm/s) in the presence of additional 1 nM Alexa647-EB3 (magenta). (f) mGFP-EB3 dwell time distribution. Number of microtubules analyzed - 548, number of mGFP-EB3 binding events - 2425. (g) Local mGFP-EB3 dwell time distributions. Dashed cyan lines are mono-exponential fits. Experimental data as in (f). (h) Local mean EB3 dwell times in the wildtype microtubule plus end region. Filled symbols correspond to the local mean dwell times calculated based on data in (g). The solid cyan line is a mono-exponential fit to the data with a decay length of 450 nm. (i) Schematic of high affinity EB3 binding to the GTP lattice of a GTPase-deficient microtubule. (j) Schematic of a growing end of a GTPase-competent microtubule displaying a gradually decreasing EB3 binding affinity (illustrated by a color gradient) as the lattice conformation changes as a consequence of GTP hydrolysis.

The online version of this article includes the following figure supplement(s) for figure 4:

**Figure supplement 1.** Flow chart of single molecule localization and spatially-resolved dwell time analysis.

**Figure supplement 2.** Spatially-resolved EB3 dwell time distributions at the ends of E254D and wildtype microtubules.

**Figure supplement 3.** Wildtype and E254D microtubules grow with similar speeds under single molecule experiment conditions.

in their true GTP state and of the conformational transitions within the GTP cap to decipher the determinants of microtubule stability and function at atomic resolution.

## Materials and methods

### Bacteria and insect cells

Bacterial strains (*Escherichia coli* DH5α, BL21 pRil, DH10MultiBac) were grown in Luria Bertani (LB) medium in the presence of appropriate antibiotics.

*Spodoptera frugiperda* strain *Sf*21 insect cells were grown in suspension at 27˚C in Sf-900 III SFM (1x) Serum Free Medium (Gibco). High Five cells were grown in suspension at 27˚C in ExpressFive SFM media (Gibco) supplemented with 16 mM L-glutamine (Gibco). Absence of mycoplasma in insect cell cultures was confirmed regularly.

### Molecular cloning

To generate an expression construct for wild type human tubulin a pFastBacDual based vector containing insect cell codon-optimized versions of human α-tubulin TUBA1B (NP_006073.2) and β-tubulin TUBB3 (NP_006077.2) (*Ti et al., 2018*) (gift from T. Kapoor) was modified by Quickchange mutagenesis as follows. The fragment encoding for the deca-histidine tag and the alanine-proline linker was removed from the N-terminus of the TUBA1B. Instead, a hexa-histidine tag encoding fragment was inserted into the internal acetylation loop of the TUBA1B between isoleucine 42 and glycine 43, a strategy used previously by *Sirajuddin et al. (2014)*; *Vemu et al. (2016)*. This resulted in the following expression construct: TUBA1B-intHis$_6$ TUBB3-Gly$_2$-Ser-Gly$_2$-TEVsite-StrepTagII (pJR374), where the StrepTagII at the C-terminus of the TUBB3 could be removed by cleavage with Tobacco Etch Virus (TEV) protease. Removal of the N-terminal tag was necessary to improve the polymerization competency of the recombinant tubulin. The new construct was subjected to further Quickchange mutagenesis of TUBA1B to produce expression constructs for GTP hydrolysis deficient (TUBA1B$^{E254A}$-intHis6 TUBB3-Gly$_2$-Ser-Gly$_2$-TEVsite-StrepTagII, pJR375) and GTP hydrolysis compromised (TUBA1B$^{E254D}$-intHis$_6$ TUBB3-Gly$_2$-Ser-Gly$_2$-TEVsite-StrepTagII, pJR376) versions of human tubulin dimers. These proteins are referred to as wildtype, E254A and E254D tubulin throughout the manuscript.

To generate a bacterial expression construct for mGFP-tagged human EB3, the coding sequence of full-length human EB3 and a long N-terminal linker were amplified by PCR from pET28a-His-mCherry-EB3 (*Montenegro Gouveia et al., 2010*) (gift from M. Steinmetz). This fragment was fused with a monomeric GFP (mGFP) (*Snapp et al., 2003*; *Zacharias et al., 2002*) sequence in a pETMZ vector to generate a bacterial expression construct encoding for the following fusion protein: His$_6$-Ztag-TEVsite-mGFP-linker-EB3 (from here on mGFP-EB3). The N-terminal His$_6$-Ztag could be removed by TEV protease treatment. The expression construct for SNAP-tagged human EB3 (SNAP-EB3) has been described elsewhere (*Jha et al., 2017*). All constructs were verified by DNA sequencing.

### Protein expression

Baculovirus preparation for recombinant human tubulin expression was carried out according to manufacturer's protocols (Bac-to-Bac system, Life Technologies) using *E. coli* DH10MultiBac and *Sf*21 insect cells. Baculovirus-infected insect cells (BIICs) were then frozen prior to cell lysis to generate stable viral stocks as described previously (*Wasilko et al., 2009*).

Recombinant human tubulin expression was induced by adding 6 ml of frozen BIICs per litre to a High Five insect cell culture grown to densities of ~$1.25 \times 10^6$ cells/ml. Cells were harvested 72 hr post-induction by centrifugation (15 min, 1000 *g*, 4˚C). Cell pellets were then washed with ice-cold 1x PBS, centrifuged again (15 min, 1000 *g*, 4˚C), frozen in liquid nitrogen, and stored in −80˚C.

mGFP-EB3 and SNAP-EB3 were expressed in *Escherichia coli* BL21 pRIL as described previously (*Jha et al., 2017*). Biotinylated monomeric *Drosophila melanogaster* kinesin-1 rigor mutant (Kin1$^{rigor}$) was expressed in *Escherichia coli* BL21 pRIL as described previously (*Roostalu et al., 2015*).

## Protein purification

High Five insect cell pellets from 2 liters of culture expressing human recombinant tubulin were resuspended in ice-cold lysis buffer (80 mM PIPES, 1 mM EGTA, 6 mM MgCl$_2$, 50 mM imidazole, 100 mM KCl, 2 mM GTP, 1 mM 2-mercaptoethanol (2-ME), pH 7.2) supplemented with protease inhibitors (Roche), DNase I (10 µg ml/ml, Sigma), using the same volume of buffer as the cell pellet. Resuspended cells were lysed by dounce homogenization (60 strokes). The lysate was then diluted 4-fold with dilution buffer (80 mM PIPES, 1 mM EGTA, 6 mM MgCl$_2$, 50 mM imidazole, 2 mM GTP, 1 mM 2-ME, pH 7.2) and clarified by ultracentrifugation (158,420x $g$, 1 hr, 4°C). The supernatant was passed through a 5 ml HisTrap HP column (GE Healthcare). The column was first washed with five column volumes (CVs) of lysis buffer, then with 5 CVs of Ni wash buffer 1 (80 mM PIPES, 1 mM EGTA, 11 mM MgCl$_2$, 2 mM GTP, 5 mM ATP, 1 mM 2-ME, pH 7.2), then with 5 CVs of Ni wash buffer 2 (80 mM PIPES, 1 mM EGTA, 5 mM MgCl$_2$, 0.1% (vol/vol) Tween-20, 10% (w/vol) glycerol), 2 mM GTP, 1 mM 2-ME, pH 7.2), and then again with 5 CVs of lysis buffer. The protein was eluted with Ni elution buffer (80 mM PIPES, 1 mM EGTA, 5 mM MgCl$_2$, 500 mM imidazole 2 mM GTP, 1 mM 2-ME, pH 7.2). The eluate was immediately diluted 6-fold with Strep binding buffer (80 mM PIPES, 1 mM EGTA, 5 mM MgCl$_2$, 2 mM GTP, 1 mM 2-ME, pH 7.2) and passed through serially con- nected 1 ml HiPrep SP FF and 5 ml StrepTag HP columns (both GE Healthcare) equilibrated in Strep binding buffer. The columns were then washed with 2 CVs of Strep binding buffer. The protein was eluted in Strep elution buffer (80 mM PIPES, 1 mM EGTA, 4 mM MgCl$_2$, 2.5 mM D-desthiobiotin, 50 mM imidazole, 2 mM GTP, 1 mM 2-ME, pH 7.2). The eluate was diluted 2-fold in Strep elution buffer, transferred on ice and incubated with TEV protease for 2 hr to remove the C-terminal Strep- TagII from TUBB3. Following TEV cleavage the eluate was centrifuged at 204,428 $g$, 10 min, 4°C. The supernatant was then passed through a further 1 ml HiPrep SP FF column equilibrated in SP wash buffer (80 mM PIPES, 1 mM EGTA, 5 mM MgCl$_2$, 2 mM GTP, pH 6.8) to remove TEV protease. Finally, the flow through now containing the purified recombinant human tubulin was passed through two serially connected pre-equilibrated HiPrep Desalting columns (GE Healthcare) to exchange the buffer to tubulin storage buffer (80 mM PIPES, 1 mM EGTA 1 mM MgCl$_2$, 0.2 mM GTP, pH 6.8). The tubulin containing fractions were pooled, concentrated (Vivaspin 30,000 MWCO, Sartorius) to above 3.5 mg/ml, ultracentrifuged (278,088 $g$, 10 min, 4°C), aliquoted, snap frozen, and stored in liquid nitrogen until use. Mass spectroscopy demonstrated that the purified human tubulin was free of insect cell tubulin (*Figure 1—figure supplement 1d*).

mGFP-EB3 and SNAP-EB3 were purified and SNAP-EB3 was labeled with SNAP-Surface-Alexa- Fluor647 (NEB) as described (referred to as Alexa647-EB3 throughout the manuscript) (*Jha et al., 2017*). Biotinylated monomeric *Drosophila melanogaster* kinesin-1 rigor mutant (Kin1[rigor]) was puri- fied as described previously (*Roostalu et al., 2015*).

Porcine brain tubulin was purified following a published protocol (*Castoldi and Popov, 2003*). Purified porcine brain tubulin was recycled and labeled with Alexa647-*N*-hydroxysuccinimide ester (NHS; Sigma-Aldrich), CF640R-NHS (Sigma-Aldrich), Atto565-NHS (Sigma-Alrich) or biotin-NHS (Thermo Scientific), as described previously (*Hyman et al., 1991*).

EB3 concentration was determined by Bradford assay, values refer to monomer concentration. Tubulin concentration was determined by UV/VIS spectroscopy measurements (absorption at 280 nm). Values refer to tubulin dimer concentration.

## Determination of microtubule nucleotide content

To determine their nucleotide content, microtubules were polymerized from different types of tubu- lin in the presence of a low concentration of short microtubule 'seeds' to initiate microtubule growth.

First microtubule 'seeds' were polymerized from 15 µM recycled porcine brain tubulin in the pres- ence of 0.5 mM GMPCPP (Jena Bioscience) in BRB80 (80 mM PIPES, 1 mM EGTA, 1 mM MgCl$_2$, pH 6.8) at 37°C for 50 min. The seeds were centrifuged at 17,000 $g$ at room temperature for 10 min, washed in warm BRB80, centrifuged at 17,000 g at room temperature, and resuspended in BRB80 to an estimated concentration of 15 µM tubulin assuming 100% nucleation and polymerization efficiency.

160 µg of purified porcine brain tubulin, human wildtype tubulin, human E254A tubulin was then used for each polymerization reaction. The polymerization reaction consisted of 20 µM tubulin

diluted in BRB80 and mixed with 26% glycerol (vol/vol) and 1 mM GTP (final concentrations). These reactions were first incubated on ice for 5 min, then transferred to 37°C, and after 1 min supplemented with 0.15 µM GMPCPP-stabilized 'seeds', and polymerized for 50 min. The samples were then centrifuged at 278,088 x *g* for 10 min at 37°C. The supernatants were aspirated, the pellets containing microtubules washed twice with 2x sample volume of warm BRB80, and resuspended in 10 mM Tris-HCl (pH 7.5). Nucleotides were then extracted by addition of 0.7% (w/vol) HClO$_4$ followed by immediate neutralization of the reaction by addition of Na-acetate to 200 mM (final concentrations). The supernatants containing the extracted nucleotides were separated from precipitated protein by centrifugation at 15,000 *g* for 5 min at 4°C and subsequently filtered through a 0.22 µm centrifugal filter (Durapore-PVDF, Millipore).

The samples (55 µl) were applied to a Zorbax SB-C18 column (4.6 × 250 mm, 5 µm pore size, Agilent Technologies) mounted on a Jasco HPLC system controlled by Chromnav software (v1.19, Jasco). The column temperature was maintained at 30°C. Nucleotides were separated under isocratic conditions in 100 mM K$_2$HPO$_4$/KH$_2$PO$_4$ (pH 6.5), 10 mM tetrabutylammonium bromide, 7% (vol/vol) acetonitrile. The absorbance was monitored using a Jasco MD-2010 Plus multi-wavelength detector. The GDP and GTP peaks were identified from comparison to the retention times of pure nucleotide standards (*Figure 1c*). The relative amounts of GDP and GTP were determined by integrating the peaks in the 260 nm absorbance channel using Chromnav software (*Figure 1d*).

## Mass spectrometry

Protein molecular mass was determined using a microTOFQ electrospray mass spectrometer (Bruker Daltonics) (*Figure 1—figure supplement 1d*). Proteins were first desalted using a 2 mm x 10 mm guard column (Upchurch Scientific) packed with Poros R2 resin (Perspective Biosystems). Protein was injected via a syringe onto the column in 10% acetonitrile, 0.10% acetic acid, washed with the same solvent and eluted in 60% acetonitrile, 0.1% acetic acid. Desalted protein was then infused into the mass spectrometer at 3 µl/min using an electrospray voltage of 4.5kV. Mass spectra were deconvolved using maximum entropy software (Bruker Daltonics).

## In vitro assays

Flow chambers were assembled similarly for all microscopy assays from poly-(L-lysine)-polyethylene glycol (PEG) (SuSoS) treated counter glass and passivated biotin-PEG-functionalized coverslips as described previously (*Bieling et al., 2010*).

### Microtubule dynamics assay

GMPCPP-stabilized biotinylated non-fluorescent microtubule 'seeds' (for iSCAT microscopy) or fluorescent seeds (containing 12% of CF640R-, or Atto565-labeled porcine brain tubulin; for TIRF microscopy) were prepared as described previously (*Bieling et al., 2010*; *Roostalu et al., 2015*). Microtubule dynamic assays were performed as detailed earlier (*Bieling et al., 2010*; *Roostalu et al., 2015*). The passivated flow chambers were incubated for 5 min with 5% Pluronic F-127 (Sigma-Aldrich) in MQ water at room temperature, washed with assay buffer (80 mM PIPES, 1 mM EGTA, 1 mM MgCl$_2$, 30 mM KCl, 1 mM GTP, 5 mM 2-ME, 0.15% (w/vol) methylcellulose (4000 cP, Sigma-Aldrich), 1% (w/vol) glucose, pH 6.8), then with assay buffer containing κ-casein (50 µg/ml, Sigma-Aldrich). Flow chambers were then incubated on a metal block on ice in κ-casein-containing assay buffer supplemented with NeutrAvidin (50 µg/ml, Life Technologies) to coat the functionalized glass surface and washed again with assay buffer. Microtubule seeds were diluted in assay buffer, flowed into the chamber and incubated for 3 min at room temperature to allow them to attach to the NeutrAvidin-coated functionalized glass surface. The chamber was then washed again with assay buffer followed by flowing in the final assay mix. The final assay mix for the microtubule dynamics assays consisted of 98% of assay buffer containing unlabeled porcine brain, wildtype or mutant recombinant human tubulin (concentrations as indicated: 5–19 µM for different tubulins) and oxygen scavengers (180 µg/ml catalase (Sigma-Aldrich), 752 µg/ml glucose oxidase (Serva)). The final assay mix also contained 2% EB3 storage buffer (50 mM Na-phosphate, 400 mM KCl, 5 mM MgCl$_2$, 0.5 mM 2-ME, pH 7.2) optionally supplemented with mGFP-EB3 (concentrations as indicated: 0–500 nM). The flow chamber was then sealed with silicone grease. Imaging was started 1 min after transferring the sample to the microscope chamber at 30°C.

## Microtubule nucleation assay

Microtubule nucleation assays (*Figure 1g-h*) were performed as described previously (*Roostalu et al., 2015*). The initial experimental steps to treat the flow cell were the same as in the microtubule dynamics assay. However, instead of NeutrAvidin the surface of the flow chamber was coated with biotinylated *Drosophila melanogaster* Kin1[rigor] mutant (500 nM) by incubation in the assay buffer on a metal block on ice for 10 min. No microtubule 'seeds' were used. Following washes with the assay buffer, the final assay mix comprising 98% of assay buffer containing unlabeled wild-type or E254A mutant recombinant tubulin (concentrations indicated) and oxygen scavengers (180 μg/ml catalase, 752 μg/ml glucose oxidase) and 2% of EB3 storage buffer was flowed into the chamber.

## Experiments with GTP-bound E254A microtubules and GMPCPP-bound wildtype or E254A microtubules

For preparation of GMPCPP-stabilized wildtype microtubules, the buffer of wildtype tubulin was first exchanged from tubulin storage buffer to BRB80. Microtubules were then polymerized from 12.5 μM wildtype tubulin in the presence of 0.5 mM GMPCPP in BRB80 at 37°C for 1 hr. The microtubules were then centrifuged at 17,000 g at room temperature for 10 min, washed in warm BRB80, centrifuged at 17,000 g at room temperature, and resuspended in BRB80. E254A microtubules were polymerized under the same conditions except in the presence of GTP (1 mM) instead of GMPCPP. These GTP-bound E254A microtubules remained stable throughout the day. To test the effect of GMPCPP on mGFP-EB3 binding to E254A microtubules, the E254A tubulin was buffer exchanged from GTP-containing storage buffer to BRB80 containing 0.5 mM GMPCPP using ZebaSpin Desalting Columns (ThermoScientific) and allowed to equilibrate for 30 min in ice. This step was repeated twice to ensure complete nucleotide exchange. After the second buffer exchange, this E254A tubulin polymerized in the presence of 0.5 mM GMPCPP at 37°C for 1 hr as described above.

The initial sample preparation for imaging and flow cell treatment were identical to the nucleation assay, except that now either stable GMPCPP-stabilized wildtype, GTP-bound E254A or GMPCPP-bound E254A microtubules were attached to the Kin1[rigor]-surface by 3 min incubation at room temperature. The chamber was then washed with assay buffer to remove unbound microtubules followed by flowing in the final assay mix consisting of 98% of assay buffer with oxygen scavengers (180 μg/ml catalase, 752 μg/ml glucose oxidase) and 2% EB3 storage buffer optionally supplemented with 20 nM or 25 nM mGFP-EB3. The experiment was performed in a similar manner when comparing the binding of 500 nM mGFP-TPX2[micro] (*Zhang et al., 2017*) to a mixture of fluorescently labeled GMPCPP-stabilized porcine brain tubulin microtubules and GTP-bound unlabeled E254A microtubules.

To determine the binding affinity of mGFP-EB3 for E254A microtubules (*Figure 2i*), E254A microtubules were first polymerized at 1 μM E254 tubulin from 0.5 μM GMPCPP-stabilized biotinylated fluorescently labeled porcine brain microtubule 'seeds' in BRB80 for 1 at 37°C for 1 hr. These microtubules were then transferred to room temperature and stored until use. The initial sample preparation steps were identical to the dynamic microtubule assay, except that now the GMPCPP-stabilized 'seeds' with stable unlabeled E254A-lattice extensions diluted in BRB80 were attached to the NeutrAvidin-coated functionalized glass surface. The final assay mix in these experiments consisted of 98% of assay buffer containing oxygen scavengers (180 μg/ml catalase, 752 μg/ml glucose oxidase) and 2% EB3 storage buffer supplemented with mGFP-EB3 (concentrations as indicated: 1–200 nM).

Samples for EB3 washout experiments with E254A microtubules (*Figure 2—figure supplement 1f*) were prepared identically to the samples for mGFP-EB3 binding affinity determination. The mGFP-EB3 concentration was 2.5 nM. After initial imaging the sample was removed from the microscope objective and washed with ~20 flow chamber volumes of assay buffer. Then assay buffer containing oxygen scavengers but no mGFP-EB3 was flowed in the chamber, and the sample was imaged again to visualize the remaining mGFP-EB3 binding.

## Single molecule assays

Single molecule experiments on dynamic microtubules were performed similarly to the dynamic assays (*Figure 4*). We chose wildtype and E254D tubulin concentration, at which wildtype and E254D microtubules grew at nearly identical growth speeds (*Figure 4—figure supplement 3*). The

final assay mix consisted of 98% of assay buffer containing wildtype tubulin (19 µM) or E254D tubulin (10 µM) and oxygen scavengers (180 µg/ml catalase, 752 µg/ml glucose oxidase) and 2% of EB3 storage buffer supplemented with 25 pM mGFP-EB3 and 1 nM Alexa647-EB3.

Single molecule experiments with E254A microtubules were carried out under identical experimental and imaging conditions as the single molecule experiments with wildtype and E254D microtubules, except using E254A microtubules grown from GMPCPP-stabilized 'seeds' (described above) instead of dynamic microtubules. The final assay mix did not include soluble tubulin but only mGFP-EB3 (10 pM) and Alexa647-EB3 (1 nM).

## Steady state bleaching assay

Samples for steady state bleaching assays were prepared identically to the single molecule experiments for each respective tubulin species (wildtype, E254D, E254A) except that only 2.5 nM mGFP-EB3 was included in the final assay mix (and no Alexa647-EB3).

# Microscopy

## Simultaneous iSCAT and TIRF microscopy

A total internal reflection fluorescence (TIRF) microscope system (Cairn Research, UK), was modified to allow simultaneous TIRF and interferometric scattering (iSCAT) microscopy (*Ortega Arroyo et al., 2016*) for the detection of GFP-EB3 and unlabeled microtubules, respectively. The collimated beams from a 488 nm and 561 nm diode laser were focused on the back focal plane of a 60 × 1.49 N.A. TIRF objective (Nikon, Japan): the 488 nm beam was positioned to give TIRF illumination and the 561 beam was positioned at ~18˚ to give epi illumination. Both beams were rapidly scanned azimuthally using a galvo scanning system (iLas2, Gataca Systems, France) to provide uniform illumination and reduce interference effects. A quadband dichroic mirror (Chroma: ZT405/488/561/640) and band-pass filters (Chroma: ET525/50, ZET561/10) were used to separate detected fluorescence and scattered 561 nm laser light. Both channels were recorded simultaneously using two EMCCD cameras (Ixon Ultra 888, Andor, UK). Both camera relays had a total magnification of 2x. Time-lapse movies were acquired with 63 ms exposure time at 1 Hz for 10 min.

For each sample, an average static background image was created by translating the sample rapidly while acquiring a movie stream, and averaging the resulting image series. The raw iSCAT movie was then divided by the corresponding average background image to produce a pseudo-flat-field corrected, normalized movie. This was then filtered using a mask in Fourier-space, to remove large-scale dynamic interference effects, and a Kalman stack filter to reduce noise.

## TIRF microscopy

For 'TIRF microscopy only' experiments (*Figures 2c-e, i–k, 3g, i-j* and *4*, *Figure 2—figure supplement 1d–g*, *Figure 4—figure supplement 2, 3*) movies were acquired using a custom TIRF microscope (iMIC, FEI Munich) at 30˚C, described in detail previously (*Maurer et al., 2014*; *Roostalu et al., 2015*). Exposure times were between 55 ms (single molecule imaging) and 150 ms (nucleation assays). Images were acquired at intervals of every 60 ms (single-molecule imaging), 1 s (nucleation assays), or 200 ms (comet analysis) keeping imaging conditions consistent for each set of experiments. For steady-state bleaching, movies were acquired with identical laser illumination powers and exposure times (55 ms), and increasing frame intervals (60, 100, 200 ms).

# Data analysis

## Image analysis

Images were processed and analyzed using the Fiji package of ImageJ (https://fiji.sc), using custom macros. Further analysis and data processing was carried out in MATLAB (MathWorks USA) and Origin (OriginLab, USA).

Kymographs were generated as described previously (*Jha et al., 2017*; *Zhang et al., 2017*). After acquisition, movies were corrected for drift when necessary, then individual microtubule growth trajectories were drawn on a maximum intensity projection of the entire movie stack (*Figure 4—figure supplement 1a*). Kymographs were generated from the movie stack along each trajectory line; subsequently, the microtubule end position was traced manually on each kymograph (*Figure 4—figure*

*supplement 1b*) (*3*). Periods including overlapping microtubules were excluded from the analysis. Microtubule growth speeds were calculated from the marked end position on the kymographs.

Our iSCAT microscopy setup allowed us to image microtubules consisting entirely of unlabeled recombinant tubulin, but did not produce images of the quality required for automated microtubule end tracking with sub-pixel precision (*Bohner et al., 2016*; *Ruhnow et al., 2011*). Therefore, we based our comet and single molecule analysis (see below) entirely on TIRF microscopy movies of fluorescently labeled EB3. For comet analysis, average spatial mGFP-EB3 intensity profiles (comets) (*Figure 3i–j*) were generated from kymographs as described previously (*Jha et al., 2017*; *Zhang et al., 2017*). Each kymograph was straightened and re-centered using the marked microtubule end positions, then aligned and averaged with other straightened kymographs. The intensities were averaged for all time points at each position along the resulting average kymograph, giving a time-averaged spatial intensity profile. The spatial resolution of this method was 1–2 pixels, that is ~200 nm. Kinetic rate constants were extracted from an exponentially modified Gaussian fit to the comet profile, as described previously (*Maurer et al., 2014*).

To determine the affinity of EB3 for binding to wildtype microtubules, average maximal comet intensities were measured as a function of the mGFP-EB3 concentration (*Figure 2c*). The equilibrium dissociation constant ($K_d$) was determined from a hyperbolic fit to the data (*Figure 2d*), because EB3 was in large excess over binding sites at microtubule ends in these experiments. To determine the EB3 binding affinity for E254A microtubules, the $K_d$ was here determined from a quadratic fit

(*Figure 2i*), $I = A\left(T + E + K_d - \sqrt{(T + E + K_d)^2 - 4\,T\,E}\right)$, where I is the measured average mGFP-

EB3 fluorescence intensity on the microtubule at a given total mGFP-EB3 concentration E and the total concentration T of binding sites along the E254E microtubules in these experiments (T was measured to be ~2.1 nM), and A is an amplitude factor (free parameter).

## Single molecule dwell time analysis

To obtain dwell time distributions for the single molecule mGFP-EB3 experiments (*Figures 2j, k* and *4a–h*), kymographs were generated and the microtubule end positions traced, as described above (*Figure 4—figure supplement 1a–b*). Using the traced end position, the corresponding x-y coordinates of the microtubule end in the original movie were calculated for each frame (*Figure 4—figure supplement 1c*). These coordinates were used to create a binary mask movie that only included points on each microtubule up to the growing end in each frame. The raw image data in each frame of a movie were analyzed with a Single Molecule Localization procedure (GDSC SMLM plugin for ImageJ: https://github.com/aherbert/GDSC-SMLM (code forked at: https://github.com/elifescien-ces-publications/gdsc-smlm) this determined the coordinates of each potential single molecule with sub-resolution precision (typically ~<30 nm). The resulting localization positions were filtered in space and time using the binary mask movie, to remove all events not localized on a microtubule (*Figure 4—figure supplement 1d*). The remaining localization events were linked together in space and time throughout the whole movie using specific thresholds: only two events that occurred within a maximum separation of one pixel (120 nm) and 30 frames (1.8 s) were linked together. This allowed for slight movement of the molecules between frames (due to e.g. lattice diffusion, microtu-bule wiggling, drift) and dark frames (due to e.g. blinking). The linking parameter values were cho-sen after a careful inspection of the effects of the parameter space on the resulting individual dwell events, for a specific kymograph. The automatically identified binding events agreed well with visu-ally identified events in test subsets of the data (*Figure 4—figure supplement 1e*). For each mole-cule, the dwell time was determined by the total time over which events were linked. The position of the molecule in the first frame of the binding event relative to the nearest microtubule end was cal-culated, creating a list of dwell times with their corresponding distances from the growing microtu-bule end (*Figure 4—figure supplement 1f*). For spatially resolved dwell time distributions, binding events were binned at specific distances from the end (*Figure 4—figure supplement 1g*) and the dwell time '1 - cumulative distribution function' (survival function) calculated for each bin. Character-istic dwell times were extracted from these distributions using a mono-exponential fit.

The average dwell time measured for single mGFP-EB3 molecules bound to the growing ends of recombinant human wildtype microtubules was 880 ms (*Figure 4f*), which is comparable to previous reports of EB dwell times being in the range of 50–300 ms (*Bieling et al., 2008*; *Bieling et al., 2007*;

*Maurer et al., 2011*; *Montenegro Gouveia et al., 2010*). The longer mean dwell time observed here is probably a consequence of the lower ionic strength buffer used.

## Steady state bleaching analysis

EB3 single molecule dwell times on wildtype and E254D microtubule ends (*Figure 4*) were essentially unaffected by bleaching, because they were much shorter than the bleaching time of ~45 s (*Figure 2—figure supplement 1g*). However, single molecule dwell times on E254A microtubules were in the range of our bleaching time and were therefore determined from a steady state bleaching analysis performed at different time intervals (*Figure 2—figure supplement 1g*; *Gebhardt et al., 2013*).

We assume that molecules bound to the microtubule bleach at a rate $k_b$ and unbind at a rate $k_{off}$. We assume an excess of unbleached EB3 molecules in solution such that at steady-state the binding rate of unbleached molecules is limited by the total unbinding rate. Solving the resulting differential equations gives a relative unbleached fraction at time $t$ after illumination of

$$F(t) = \left(k_{off} + k_b e^{-k_T t}\right)/k_T$$

Where $k_T = k_b + k_{off}$.

Movies taken at different frame intervals were background corrected using a 50 pixel rolling ball subtraction. The mean intensity over the field of view was calculated for each frame, and normalized to the value in the first frame. Decay curves were fit with the function above, with $k_{off}$ shared globally (*Figure 2—figure supplement 1g*).

## Data and materials availability

All data are available in the manuscript or the supplementary materials. Correspondence regarding data and materials should be addressed to TS.

## Acknowledgements

We thank Tarun Kapoor for tubulin and Michel Steinmetz for EB3 plasmids that were used for cloning, Steve Howell from the Mass Spectrometry Proteomics Science Technology Platform and the Structural Biology Science Technology Platform of the Francis Crick Institute for support, Ivo Telley and Jamie Rickman for helpful discussions, and Peter Bieling, Franck Fourniol, Nate Goehring and Sebastian Maurer for critical reading of the manuscript.

## Additional information

### Competing interests

Thomas Surrey: Reviewing Editor, eLife. The other authors declare that no competing interests exist.

### Funding

| Funder | Grant reference number | Author |
| --- | --- | --- |
| Cancer Research UK | FC001163 | Johanna Roostalu<br>Claire Thomas<br>Nicholas Ian Cade<br>Thomas Surrey |
| Medical Research Council | FC001163 | Johanna Roostalu<br>Claire Thomas<br>Nicholas Ian Cade<br>Thomas Surrey |
| Wellcome Trust | FC001163 | Johanna Roostalu<br>Claire Thomas<br>Nicholas Ian Cade<br>Thomas Surrey |
| Wellcome Trust | 100145/Z/12/Z | Johanna Roostalu |

| European Research Council | Advanced Grant 323042 | Thomas Surrey |
| Spanish Ministry of Economy and Competitiveness | | Thomas Surrey |
| Centro de Excelencia Severo Ochoa | | Thomas Surrey |
| CERCA Programme/Generalitat de Catalunya | | Thomas Surrey |

The funders had no role in study design, data collection and interpretation, or the decision to submit the work for publication.

### Author contributions

Johanna Roostalu, Conceptualization, Resources, Formal analysis, Funding acquisition, Validation, Investigation, Visualization, Methodology, Project administration, Supervision; Claire Thomas, Conceptualization, Resources, Validation, Investigation, Methodology; Nicholas Ian Cade, Conceptualization, Resources, Software, Formal analysis, Validation, Investigation, Visualization, Methodology; Simone Kunzelmann, Ian A Taylor, Investigation, Methodology; Thomas Surrey, Conceptualization, Resources, Supervision, Funding acquisition, Validation, Project administration

### Author ORCIDs

Johanna Roostalu https://orcid.org/0000-0002-6757-0468
Simone Kunzelmann http://orcid.org/0000-0002-2678-0549
Ian A Taylor https://orcid.org/0000-0002-6763-3852
Thomas Surrey https://orcid.org/0000-0001-9082-1870

### Decision letter and Author response

Decision letter https://doi.org/10.7554/eLife.51992.sa1
Author response https://doi.org/10.7554/eLife.51992.sa2

## Additional files

### Supplementary files

• Transparent reporting form

### Data availability

All data generated or analysed during this study are included in the manuscript and supporting files.

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
