## [Decision Letter]

**Acceptance summary:**

This carefully executed in vitro reconstitution study resolves some important questions concerning the nature of the cap that stabilizes growing microtubules. The study will be of interest to a broad biological audience interested in cytoskeletal regulation.

**Decision letter after peer review:**

Thank you for submitting your article "The speed of GTP hydrolysis determines GTP cap size and controls microtubule stability" for consideration by *eLife*. Your article has been reviewed by three peer reviewers, and the evaluation has been overseen by Anna Akhmanova as the Senior and Reviewing Editor. The reviewers have opted to remain anonymous.

The reviewers have discussed the reviews with one another and the Reviewing Editor has drafted this decision to help you prepare a revised submission.

The reviewers agreed that this is a rigorously performed study, which addresses important questions about the relationship between the stabilizing cap at growing microtubule ends, the nucleotide state of tubulin and the tubulin state recognized by End Binding proteins. The finding that EB3 specifically recognizes the GTP form of tubulin is an important, though perhaps not entirely surprising contribution to the field. Further, all three reviewers agreed that the description of a gradient of EB binding affinity along the microtubule end is the most novel and interesting part of the paper, but also the part that needs to be worked out better. Specifically, the conformational transition model raised a number of critiques, and these should be addressed by additional experiments and analyses,and possibly also by adjusting the writing of the paper. Since there might be different ways of extending this part of the paper, I include below a summary of the reviewers' discussions and also the full reviews.

The reviewers felt that you should be more clear about what you mean by 'conformation' – do you mean 'expanded/compacted' microtubule lattices, as described by structural work from Nogales lab? Could GTP tubulin exist in either the expanded or the compacted state according to a conformational equilibrium? To begin to address this question, it would be great to get data on whether E254A/D lattices are expanded or compacted. It should be possible to obtain these data without generating a high resolution structure, perhaps by using TPX2 binding experiments or by a 2D cryo-EM analysis.

Further, the conclusions about a conformational gradient are based on a limited set of data, and as, pointed out by the reviewers, alternative explanations of the obtained results are possible. One could consider performing experiments with a monomeric EB3 protein, because they might help to rule out or provide support for the model proposed by reviewer 1. Another idea raised during the consultation between the reviewers was to use as a binding substrate microtubules grown from mixtures of E254A and wild type tubulin at different ratios. Two reviewers questioned the assignment of 'mono exponential' distributions in Figure 4, and this critique should be addressed. Finally, two reviewers found that the split comets deserve some analysis, and the faster growth rate displayed by E254D microtubules requires some attention.

Reviewer #1:

Roostalu et al. have successfully created GTPase-dead and GTP-slow recombinant human tubulin, which is a significant technical breakthrough. They use this new tubulin, along with fluorescent-EB3, to probe the behavior the GTP cap. They make the surprising observation that the dwell time of EB3 molecules changes as you go deeper in the cap. As expected from the Surrey lab, this is a high quality paper addressing a central issue in the microtubule field, namely the role of GTP hydrolysis.

Major comments:

1) The dwell time of EB3 on the microtubule decreases as you move deeper into the cap. The decrease in dwell times is hypothesized to be caused by a gradual "conformational gradient" that reduces EB3 affinity. The visual representation of this idea is the color gradient from red to yellow in the schematic in Figure 4J.

I have an alternative hypothesis for why the dwell times decrease. Consider that EB3 is an "obligatory dimer" (Sen et al., 2013), so its dissociation requires both CH domains to unbind from their respective binding sites. At the very end of the microtubule, EB3 is likely to find two adjacent sites where all of the relevant tubulin dimers are in the GTP state. (The site has 4 dimers, but we can leave aside for the moment the question of their relative relevance.) As you go deeper into the cap, the binding sites will become a mixture of GTP, GDP-Pi, and GDP states. These states have different affinities for EB3. In some cases, a single, high-affinity, GTP site may be adjacent only to GDP sites. In the context of the EB3 dimer, the site mismatch means that one CH domain will bind a GTP site and one can only bind a low-affinity GDP site. EB3 becomes functionally monomeric, hanging on to the microtubule with only one hand. In contrast, EB3 at the very end of the microtubule is holding on with both hands. Note: the EB3 construct they are using appears to be full-length when I trace back through their Materials and methods references.

Can this alternative framework explain the decrease in dwell times with depth inside the cap (Figure 4C)? More specifically: a gradual transition from dimeric to monomeric affinity conditions, driven by the probabilistic availability to two binding sites with different affinities? Are the measurements precise enough to rule out this hypothesis? The visual representation of this idea would NOT be a color gradient but rather an increased "speckling" of red and yellow blocks.

2) I have some technical questions about the "spatially-resolved EB3 dwell time distributions". They divide the microtubules into bins based on distance from the plus-end. The bin size is approximately 0.6 μm for the E254D microtubules (see legend to Figure 4C, see Figure 4—figure supplement 2) and 0.2 μm for the wildtype microtubules (Figure 4—figure supplement 2). These bins are relatively small when you consider that: (1) the microtubule end-position is not determined with sub-pixel accuracy, but is determined rather using the "traced end position" from a manual-tracing of a kymograph, which they state has 0.2 μm accuracy, (2) there is error in the single-molecule localization of the EB3 of 30 nm, and (3) the microtubule is growing at 70 nm/s, which means that an EB3 molecule could start its residence time in one bin and yet end it in another. The first two points make me uncertain whether the molecules are being correctly placed into their bins. The 3rd point is more conceptual: what is the best way to treat a molecule whose bin changes beneath it?

3) How mono-exponential are the spatially-resolved dwell time distributions (Figure 4C)? They appear to start off linear but then deviate from linearity at, e.g., t = 4 s for the 1.47 μm bin.

The central argument of the paper hinges on the fact that Figure 4B is not mono-exponential but Figure 4C is mono-exponential. Are the fits really strong enough to support this? The distributions are described as "strikingly mono-exponential". What does strikingly mean in terms of goodness of fit?

4) The split-comet phenomenon is mentioned briefly, along with their "curved appearance" – but there is very little quantification of the behavior. How often are they observed? How bright are the split comets compared to a full comet, etc.

5) The tubulin that they use retains an internal His tag on the α subunit. That's fine; the Roll-Mecak lab uses similar constructs. The Materials and methods are clear about the retention of the internal His tag but the main text is not. I think it's important to be clear throughout.

A suggestion:

The E254A and E254D microtubules are characterized in terms of their EB3 affinities. Perhaps the authors could explore the binding preference of TPX2, as it would report on the expansion/compaction state of these lattices? Alternatively, in the concluding paragraph, the authors say that high-resolution structures are on the way. But one doesn't necessarily need a 3.4 A structure to answer the most important question: are these lattices expanded or compacted?

Notes on the writing:

– The hypothesis that a nucleotide gradient "might translate into a conformational stability gradient" comes out of nowhere. What motivates this hypothesis? Are there data, structures, kinetic measurements, conceptual arguments, etc, that would motivate this idea?

– The concluding paragraph suggests that the "normal" GTP hydrolysis rate might be under evolutionary pressure. Are there measurements of the GTPase activity with non-human proteins that would provide support for this idea?

Reviewer #2:

This is an interesting and well-executed paper that addresses interesting questions about the microtubule's stabilizing cap, how it relates to nucleotide state, and what is the preferred state that EB proteins recognize. The approach was to purify site-directed recombinant human α/β-tubulins with mutations at a putative GTPase residue. The experiments are performed rigorously and for the most part described clearly, and the work has been done to a high technical standard.

The main findings of the paper are: (i) that abolishing (or at least substantially slowing; E254A) GTPase results in very stable microtubules akin to GMPCPP microtubules, (ii) that moderately slowing (E254D) GTPase results in microtubules with longer EB comets that are also more stable, and (iii) that in these longer comets (and also wild-type comets), there appears to be a gradient of EB binding affinity, with the highest affinity being closest to the growing end. Although it has not been shown directly before using a mutant, I did not find it all surprising that reducing GTPase rate increased microtubule stability, but it is nice to see this in the way that it is shown here. The findings involving EB binding are more interesting: they provide some of the most direct support for the idea of an adaptable microtubule lattice, and they raise questions about what states commonly used GTP analogs are giving.

Major comments:

– I wonder if the balance of the manuscript should change somewhat if to be published in *eLife*. I think the site-specific EB analysis is probably the most interesting and mechanistic part of the manuscript, and the authors might want to place a little more emphasis there (and place less emphasis on some of the obvious-sounding claims). I don't think new experiments are required, but they may want to go deeper into some of the analysis of the EB binding.

– The authors observe that EB3 does not bind to recombinant GMPCPP microtubules, even though in other ways GMPCPP has been taken to be a good GTP analog. The authors did not mention that GMPCPP generally gives 14 protofilament microtubules. Could the authors commend on whether they think the switch in protofilament number or the different conformation of tubulin (or both) account for the lack of EB3 binding?

– Some of the language suggests there is a 1:1 correspondance between nucleotide state and lattice state, but their results indicate the opposite. I think this should be made more clear.

– It seems E254D microtubules grow ~twice as fast as wild-type. This is unexpected, and the authors really don't say much about it.

– In Figure 4D, the authors fit the EB dwell time decay as a function of distance from the tip. This is what they measured, of course. Since they know the growth rates, is there anything interesting to learn from plotting the decay against time (or just transforming to get the time dependence)? It seems that even speculatively they might be able to get a little more specific about the likely mix of GTP/GDP at different distances from the tip, and/or say something more quantitative about what the GTPase rate must be.

– The spatially resolved EB binding analysis was a strength of the paper for me. I think it also has the potential to be confusing to people, because the language/fits imply a series of discrete 'states' but it seems these states would have to be heterogenous in terms of nucleotide state. A few more sentences about this might be useful.

– The supplemental discussion felt looser than the rest of the paper. In particular, while the authors ascribe various deficiencies to nucleotide analogs, they do not seem to consider the possibility that the mutation(s) they made might also perturb the structure. This criticism applies to the main text also. Some version of the discussion of induced fit should probably be incorporated into the main text.

Reviewer #3:

The study by Roostalu et al. addresses the fundamental question of how GTP-tubulin at the growing tip of a microtubule can be detected by other proteins. The present study examines the binding of the major plus end tip tracking protein, EB3, and its ability to specifically recognize the GTP form of tubulin. In recent years, it has been controversial as to whether tip trackers are recognizing GTP or possibly another nucleotide/conformational state, such as GDP-Pi-tubulin. Here, it is shown, using tubulin mutants that either fail to hydrolyze or hydrolyze slowly, that EB3 specifically recognizes the GTP form of tubulin. This is an important finding for the field, one that is achieved through elegant experimental studies of mutant tubulin and careful quantitative analysis, for which the authors are to be commended. However, in the final analysis, the authors invoke a GDP-Pi intermediate state, without strong evidence to justify it. Rather, it seems possible that a simpler alternative explanation that only assumes GTP and GDP states, as suggested by their data, is not ruled out. Thus, I am concerned that the study, while making an important contribution to the field, may be misleading in its final interpretation.

Major comments:

1) Conclusion of a GDP-Pi state based on non-exponentially distributed dwell times is problematic.

a) Figure 4C. The spatially resolved dwell time is interesting, but the two positions that are farther away from the tip appear that they may be bi-exponential. It seems the non-exponential distribution might be expected as the *k*_off_ jumps when the hydrolysis occurs, which would give spatially varying dwell times and nonexponential distributions as the hydrolysis can occur at random (Poisson process) during the observation time. So it still seems that the data could be consistent with a simple GTP-GDP model, with no need for an additional third state of GDP-Pi.

b) Figure 4J is unconvincing as the simpler model of only two nucleotide states (GTP and GDP) has not been ruled out. To rule this out it will be necessary to do model-convolution to make it convincing that the analysis method is not yielding spurious results. Even then, the EB binding could be dependent on the local neighborhood of nucleotide state (see Seetapun et al., Figure 2—figure supplement 1 for conformational spread modeling), which seems reasonable since the binding site is at the interface between tubulins. Overall, model-convolution on the microtubule addition-loss-hydrolysis and EB binding-unbinding to assess the model is needed to rule out the simpler GTP-GDP model. Even then the argument for a GDP-Pi state is not compelling.

2) The values for KDs of 8 nM and 40 nM for EB3 binding to GTP- and GDP-tubulin, respectively, seem very strong compared to Seetapun et al. values of 3.8 µM and 55 µM, respectively, in vivo. Why is this, and are the in vitro results informative of the tip tracking in living cells?

3) How big is the cap with wildtype tubulin in vitro? Note: "cap size" is in the title, but it is not estimated, despite a lot of nice quantitation. Previous estimates put it at ~40 in vitro (at 5 µM wildtype tubulin) and ~750 in vivo (Schek et al., 2006; Seetapun et al., 2012), a disparity that is largely explained by the relative disparity in the net growth rates under these two conditions. These papers should be cited, as previous estimates of cap size. (Note: need to account for tubulin concentration, e.g. 19 µM wildtype tubulin in Figure 3J vs. 5 µM in Schek et al. and 7 µM in. Seetapun et al.).

---

## [Author Response]

The reviewers agreed that this is a rigorously performed study, which addresses important questions about the relationship between the stabilizing cap at growing microtubule ends, the nucleotide state of tubulin and the tubulin state recognized by End Binding proteins. The finding that EB3 specifically recognizes the GTP form of tubulin is an important, though perhaps not entirely surprising contribution to the field.

We thank the reviewers for their overall very positive evaluation of the quality, novelty and importance of our work. We however respectfully disagree with the view that the demonstration of EBs binding GTP microtubules is less significant, because it is perceived as not surprising. The question whether EBs bind to the GTP conformation of microtubules could not be answered with certainty in the past. Previous studies have used various nucleotide analogs to address questions of the nucleotide state. However, EBs bind with quite different affinities to microtubules grown in either GMPCPP or GTPγS, which has left room for different interpretations and resulted in lack of clarity. In our Introduction, we now provide more background and explain these different interpretations that arose from partly contradictory observations. We believe that this helps to provide a better context and highlights the value of our experiments with GTP microtubules. In our view, we present here for the first time evidence that EBs indeed bind the GTP state of microtubules with high affinity, providing clarity concerning a central question about microtubule biochemistry that has remained unsolved and has been a matter of speculation.

Further, all three reviewers agreed that the description of a gradient of EB binding affinity along the microtubule end is the most novel and interesting part of the paper, but also the part that needs to be worked out better. Specifically, the conformational transition model raised a number of critiques, and these should be addressed by additional experiments and analyses,and possibly also by adjusting the writing of the paper. Since there might be different ways of extending this part of the paper, I include below a summary of the reviewers' discussions and also the full reviews.

We have performed additional analysis (going beyond what the reviewers asked) which helped us to improve the clarity of the description of the measured affinity gradient. We describe this in detail in our response to the individual points raised by the reviewers and provide an improved description and discussion in the manuscript.

The reviewers felt that you should be more clear about what you mean by 'conformation' – do you mean 'expanded/compacted' microtubule lattices, as described by structural work from Nogales lab? Could GTP tubulin exist in either the expanded or the compacted state according to a conformational equilibrium? To begin to address this question, it would be great to get data on whether E254A/D lattices are expanded or compacted. It should be possible to obtain these data without generating a high resolution structure, perhaps by using TPX2 binding experiments or by a 2D cryo-EM analysis.

EBs sense the conformation of their binding site. And the conformation of this binding site is affected by the nucleotide state. We make this clear in the revised Discussion. Fluorescence microscopy experiments cannot visualise the degree of lattice 'compaction' or 'expansion' that can be observed by cryo-electron microscopy. Therefore, we do not make statements about such lattice characteristics. Following a suggestion of a reviewer, we added experiments with a fragment of TPX2 to the manuscript but consider electron microscopy experiments beyond the scope of this already extensive study.

Further, the conclusions about a conformational gradient are based on a limited set of data, and as, pointed out by the reviewers, alternative explanations of the obtained results are possible. One could consider performing experiments with a monomeric EB3 protein, because they might help to rule out or provide support for the model proposed by reviewer 1. Another idea raised during the consultation between the reviewers was to use as a binding substrate microtubules grown from mixtures of E254A and wild type tubulin at different ratios. Two reviewers questioned the assignment of 'mono exponential' distributions in Figure 4, and this critique should be addressed. Finally, two reviewers found that the split comets deserve some analysis, and the faster growth rate displayed by E254D microtubules requires some attention.

We respectfully disagree with the view that our data set is too limited. Not many labs are currently able to make high quality recombinant tubulin. Although it cannot be produced in amounts as large as for animal brain tubulin, our single molecule imaging data sets here are larger than previous data sets, even when compared to experiments made with brain tubulin. Otherwise our new spatially resolved dwell time analysis at growing microtubule ends would not have been possible. We consider experiments with monomeric EB and mixed microtubule lattices beyond scope, because they have their own challenges and become easily studies in their own right. We have addressed the issue of the 'mono-exponential distributions' by additional analysis. We agree that the increased growth speed of the E254D mutant is interesting as well as the observation of ‘split comets’, which supports our conclusions about enhanced microtubule stability. However, we cannot characterize and address all the aspects that are interesting about these new mutants in such great detail (structure, kinetics, ‘split comets’, binding of different MAPs) in a single paper.

Reviewer #1:[…]1) The dwell time of EB3 on the microtubule decreases as you move deeper into the cap. The decrease in dwell times is hypothesized to be caused by a gradual "conformational gradient" that reduces EB3 affinity. The visual representation of this idea is the color gradient from red to yellow in the schematic in Figure 4J.I have an alternative hypothesis for why the dwell times decrease. Consider that EB3 is an "obligatory dimer" (Sen et al., 2013), so its dissociation requires both CH domains to unbind from their respective binding sites. At the very end of the microtubule, EB3 is likely to find two adjacent sites where all of the relevant tubulin dimers are in the GTP state. (The site has 4 dimers, but we can leave aside for the moment the question of their relative relevance.) As you go deeper into the cap, the binding sites will become a mixture of GTP, GDP-Pi, and GDP states. These states have different affinities for EB3. In some cases, a single, high-affinity, GTP site may be adjacent only to GDP sites. In the context of the EB3 dimer, the site mismatch means that one CH domain will bind a GTP site and one can only bind a low-affinity GDP site. EB3 becomes functionally monomeric, hanging on to the microtubule with only one hand. In contrast, EB3 at the very end of the microtubule is holding on with both hands. Note: the EB3 construct they are using appears to be full-length when I trace back through their Materials and methods references.Can this alternative framework explain the decrease in dwell times with depth inside the cap (Figure 4C)? More specifically: a gradual transition from dimeric to monomeric affinity conditions, driven by the probabilistic availability to two binding sites with different affinities? Are the measurements precise enough to rule out this hypothesis? The visual representation of this idea would NOT be a color gradient but rather an increased "speckling" of red and yellow blocks.

This is an interesting point and it turns out that it is important to consider whether EBs diffuse on the microtubule lattice while they are bound:

If they do not diffuse, the reviewer's model is not supported by the data. As pointed out by the reviewer, in the case of a "speckled" lattice (some GTP, some GDP, some GDP+Pi tubulins), one expects many binding affinities for a dimeric EB (which we use here). This would then result in complicated, clearly non-mono-exponential local dwell time distributions, which we do not observe.

However, if EBs can diffuse on the lattice, as reported earlier (Lopez and Valentine, 2016), they are expected to have a "mixed" affinity resulting from the various affinities of the binding sites they visit during lattice diffusion, leading again to mono-exponential local dwell time distributions. In our experiments, we observe much less diffusion than reported in Lopez and Valentine (possibly due to our lower ionic strength buffer). Performing a mean-squared-displacement (MSD) analysis of the positional variation of single EBs on E254D microtubules, we find a diffusion coefficient of D=8.4 x10^-4^ um^2^/s. This means that during a typical dwell time of a few seconds, EBs appear to diffuse a distance of ~50 nm, corresponding to several tubulins.

Therefore, our data do not allow to distinguish between the reviewer's model of a speckled nucleotide state lattice in the cap of growing microtubule ends and a model in which tubulins hypothetically change their conformation in a concerted manner.

Nevertheless, we clearly observe an affinity gradient for EB binding at growing microtubule ends, as we state in our manuscript. We explain now in our Discussion that the measured affinities are 'average' affinities integrating information of the conformation of several EB binding sites visited by EBs during lattice diffusion. We also clarify that our colour gradient in the Legend of the schematic figure is intended to illustrate affinity gradient for EB binding.

2) I have some technical questions about the "spatially-resolved EB3 dwell time distributions". They divide the microtubules into bins based on distance from the plus-end. The bin size is approximately 0.6 μm for the E254D microtubules (see legend to Figure 4C, see Figure 4—figure supplement 2A) and 0.2 μm for the wildtype microtubules (Figure 4—figure supplement 2C). These bins are relatively small when you consider that: (1) the microtubule end-position is not determined with sub-pixel accuracy, but is determined rather using the "traced end position" from a manual-tracing of a kymograph, which they state has 0.2 μm accuracy, (2) there is error in the single-molecule localization of the EB3 of 30 nm, and (3) the microtubule is growing at 70 nm/s, which means that an EB3 molecule could start its residence time in one bin and yet end it in another. The first two points make me uncertain whether the molecules are being correctly placed into their bins. The 3rd point is more conceptual: what is the best way to treat a molecule whose bin changes beneath it?

A bin size of 200 nm and 600 nm was chosen for wildtype and E254D microtubules that grew at ~60 nm/s in experiments used for spatially resolved dwell time analysis. Figures 4F and 4B show that >90% of binding events have dwell times <4s (wildtype) or <6s (E254D); hence, only a very small fraction of EBs will be at a distance from the microtubule end that corresponds then to a different bin at the end compared to the beginning of the binding event. Mis-assignment to incorrect bins due the errors in determining the microtubule end position and the EB position, could lead to incorrectly mixing up different affinities in a bin which would make dwell time distributions less mono-exponential. However, this is not observed, therefore this is not an issue here. We do observe that total dwell time distributions which include all binding events are clearly less mono-exponential (Figure 4B) than the local dwell time distributions (Figure 4C). Importantly, the measured dwell times of the local dwell time distributions quite obviously change with distance from the microtubule end (Figure 4C), demonstrating the existence of an affinity gradient and providing a very simple explanation for the non-mono-exponential nature of the dwell time distribution of all events.

3) How mono-exponential are the spatially-resolved dwell time distributions (Figure 4C)? They appear to start off linear but then deviate from linearity at, e.g., t = 4 s for the 1.47 μm bin.The central argument of the paper hinges on the fact that Figure 4B is not mono-exponential but Figure 4C is mono-exponential. Are the fits really strong enough to support this? The distributions are described as "strikingly mono-exponential". What does strikingly mean in terms of goodness of fit?

Our main conclusion of the spatially resolved dwell time analysis is that one can measure an affinity gradient for EB binding in the cap at growing microtubule ends. This is demonstrated by a change of the characteristic local dwell times with distance from the microtubule end. This result is independent of the question how well distributions are fitted by a mono-exponential distribution. However, we have demonstrated this now quantitatively: we state in the legend of Figure 4 that a mono-exponential fit to the total dwell time distribution (Figure 4B) has a reduced chi-squared value two-times larger than that of a *global* mono-exponential fit to all the distance-binned data (Figure 4—figure supplement 2A). Furthermore, it should be noted that the logarithmic display of the dwell time distributions overemphasizes the small deviations from mono-exponential behaviour in the tail of some distributions – there is remarkably little deviation over the largest decade of data.

4) The split-comet phenomenon is mentioned briefly, along with their "curved appearance" – but there is very little quantification of the behavior. How often are they observed? How bright are the split comets compared to a full comet, etc.

We clarify in the revised manuscript that comets with curved appearance occur for the E254D mutant at higher tubulin concentrations (8 µM or above), i.e. when microtubule growth speeds are fast and when considerable microtubule nucleation is observed. We found it interesting to report the 'split comets' (as previously reported in Portran et al., NCB, 2017) as they suggest that even partial tubes can grow under some conditions if caps are hyperstabilised and long. This is however not the main point of the manuscript. This could be characterized further in the future combined with structural studies.

5) The tubulin that they use retains an internal His tag on the α subunit. That's fine; the Roll-Mecak lab uses similar constructs. The Materials and methods are clear about the retention of the internal His tag but the main text is not. I think it's important to be clear throughout.

We agree. We state now in the main text that we use an internal His-tag. We already stated in the Materials and methods of the original version of the manuscript that an N-terminal tag on α-tubulin, even if it is very small, did not allow normal microtubule growth in our hands. We observed that highly purified recombinant wildtype tubulin that had 3 additional amino acids left at the N-terminus of α-tubulin after protease cleavage of an N-terminal His tag did not elongate from stabilised microtubule ‘seeds’. Therefore, we moved the His-tag to the position in α-tubulin that has been previously used by the Roll-Mecak, Vale and Rice labs for human and yeast tubulins. This allows normal microtubule growth.

A suggestion:The E254A and E254D microtubules are characterized in terms of their EB3 affinities. Perhaps the authors could explore the binding preference of TPX2, as it would report on the expansion/compaction state of these lattices? Alternatively, in the concluding paragraph, the authors say that high-resolution structures are on the way. But one doesn't necessarily need a 3.4 A structure to answer the most important question: are these lattices expanded or compacted?

We agree that electron microscopy of these mutant microtubules is very interesting, but we consider this beyond the scope of this manuscript. Instead, we have made the suggested TPX2 experiment and include it in the revised version of the manuscript (Figure 2—figure supplement 1E). We used a fragment of TPX2 that we previously called 'TPX2^micro^' and that binds preferentially to GMPCPP microtubules, but not very efficiently to growing microtubule ends (Zhang et al., 2017). We find that this construct does not bind well to GTP-containing E254A microtubules, demonstrating that the TPX2^micro^ binding site conformation is similar in the cap of normal growing microtubule ends and all along GTPase-deficient microtubules, but different on GMPCPP microtubules. Together with our findings for the EB binding preferences, this means that GTP-containing E254A microtubules mimic better the properties of normal GTP caps than GMPCPP microtubules, at least as far as EB3 and TPX2^micro^ binding is concerned.

Notes on the writing:– The hypothesis that a nucleotide gradient "might translate into a conformational stability gradient" comes out of nowhere. What motivates this hypothesis? Are there data, structures, kinetic measurements, conceptual arguments, etc, that would motivate this idea?

The idea of a stability gradient is supported by our data of GTPase-dead microtubules (E254A) being extremely stable and nucleating extremely efficiently, and the microtubules with slow GTPase hydrolysis having an intermediate stability and nucleation capacity, in between E254A and wildtype microtubules. This allows the hypothesis that a conformational gradient in the GTP cap corresponds to a stability gradient of the lattice as it transforms from the GTP to the GDP state.

– The concluding paragraph suggests that the "normal" GTP hydrolysis rate might be under evolutionary pressure. Are there measurements of the GTPase activity with non-human proteins that would provide support for this idea?

We removed the mentioning of 'evolutionary pressure'.

Reviewer #2:[…]– I wonder if the balance of the manuscript should change somewhat if to be published in eLife. I think the site-specific EB analysis is probably the most interesting and mechanistic part of the manuscript, and the authors might want to place a little more emphasis there (and place less emphasis on some of the obvious-sounding claims). I don't think new experiments are required, but they may want to go deeper into some of the analysis of the EB binding.

We have expanded our Introduction to better explain the conflicting interpretations in the literature with regards to nucleotide state and microtubule stability. We believe that these contradictions necessitate the experiments that we report here with pure GTP microtubules in order to be able to clearly answer a fundamental questions about the nature of the GTP cap. We think that all our findings and not only the final part of the study provide important new information and clarify unresolved issues about the GTP cap and microtubule dynamics.

– The authors observe that EB3 does not bind to recombinant GMPCPP microtubules, even though in other ways GMPCPP has been taken to be a good GTP analog. The authors did not mention that GMPCPP generally gives 14 protofilament microtubules. Could the authors commend on whether they think the switch in protofilament number or the different conformation of tubulin (or both) account for the lack of EB3 binding?

Although EBs have been shown to promote the formation of 13-protofilament microtubules (Vitre et al., NCB, 2008; Maurer et al., 2011; Howes et al., Cell Cycle, 2018), they do also bind to microtubules with other protofilament numbers. There is typically a mixture of microtubules with different protofilament numbers in in vitro experiments. Nevertheless, there are no strong differences in EB binding between different microtubules neither at growing microtubule ends, nor on GTPγS microtubules to which EBs bind with high affinity (e.g. Maurer et al., 2011; Roth et al., 2018)) indicating that EBs can bind well also to 14 protofilament microtubule ends and 14 protofilament GTPγS microtubules. This demonstrates that the nucleotide state plays the dominant role when considering the behaviour of EBs on GMPCPP microtubules.

– Some of the language suggests there is a 1:1 correspondance between nucleotide state and lattice state, but their results indicate the opposite. I think this should be made more clear.

Our data do not allow us to decide whether there is a 1:1 correspondence between nucleotide and lattice state of an individual tubulin. However, we observe a correlation between EB binding affinity and nucleotide state. EB binding probably samples the average lattice state of the area it can diffuse over, as we explain in our new Discussion.

– It seems E254D microtubules grow ~twice as fast as wild-type. This is unexpected, and the authors really don't say much about it.

We agree that this is unexpected. We cannot explain this observation at the moment and expect that a combination of careful kinetic and structural experiments will be required to obtain a satisfactory mechanistic explanation. We consider this a study in itself and a task for the future.

– In Figure 4D, the authors fit the EB dwell time decay as a function of distance from the tip. This is what they measured, of course. Since they know the growth rates, is there anything interesting to learn from plotting the decay against time (or just transforming to get the time dependence)? It seems that even speculatively they might be able to get a little more specific about the likely mix of GTP/GDP at different distances from the tip, and/or say something more quantitative about what the GTPase rate must be.

The rate of the decay of the dwell times over distance together with the growth speed provides the time scale of the conformational lattice change that causes the EB binding affinity to change. This most likely reflects more or less directly the time scale of GTP hydrolysis, with the caveat that we do not know how fast phosphate is released.

– The spatially resolved EB binding analysis was a strength of the paper for me. I think it also has the potential to be confusing to people, because the language/fits imply a series of discrete 'states' but it seems these states would have to be heterogenous in terms of nucleotide state. A few more sentences about this might be useful.

We have added a discussion of this issue to our expanded Discussion.

– The supplemental discussion felt looser than the rest of the paper. In particular, while the authors ascribe various deficiencies to nucleotide analogs, they do not seem to consider the possibility that the mutation(s) they made might also perturb the structure. This criticism applies to the main text also. Some version of the discussion of induced fit should probably be incorporated into the main text.

We have re-written the manuscript in a longer format, with a separate Introduction and Discussion. To address the valid concern of the mutation potentially perturbing the microtubule structure, we performed an additional experiment. We removed GTP from purified E254A tubulin by buffer exchange and grew E254A microtubules in the presence of GMPCPP. These microtubules display strongly reduced EB binding, clearly demonstrating that the EB binding affinity, and hence the conformation of the microtubule, is indeed determined by the presence of the nucleotide and not by the E254A mutation itself (Figure 2—figure supplement 1D).

Reviewer #3:The study by Roostalu et al. addresses the fundamental question of how GTP-tubulin at the growing tip of a microtubule can be detected by other proteins. The present study examines the binding of the major plus end tip tracking protein, EB3, and its ability to specifically recognize the GTP form of tubulin. In recent years, it has been controversial as to whether tip trackers are recognizing GTP or possibly another nucleotide/conformational state, such as GDP-Pi-tubulin. Here, it is shown, using tubulin mutants that either fail to hydrolyze or hydrolyze slowly, that EB3 specifically recognizes the GTP form of tubulin. This is an important finding for the field, one that is achieved through elegant experimental studies of mutant tubulin and careful quantitative analysis, for which the authors are to be commended. However, in the final analysis, the authors invoke a GDP-Pi intermediate state, without strong evidence to justify it. Rather, it seems possible that a simpler alternative explanation that only assumes GTP and GDP states, as suggested by their data, is not ruled out. Thus, I am concerned that the study, while making an important contribution to the field, may be misleading in its final interpretation.

We thank this reviewer for the overall positive assessment of the manuscript and the acknowledgement of the importance of our findings. We believe that some of our statements or the final scheme might have caused a misunderstanding. We did not intend to directly draw conclusions about the GDP-Pi intermediate state from our data as presented here but simply aimed to put our new findings about the GTP hydrolysis into the general context of the current knowledge of different nucleotide states at the microtubule end. We have changed or presentation, expanding our Discussion to clarify this point.

Major comments:1) Conclusion of a GDP-Pi state based on non-exponentially distributed dwell times is problematic.a) Figure 4C. The spatially resolved dwell time is interesting, but the two positions that are farther away from the tip appear that they may be bi-exponential. It seems the non-exponential distribution might be expected as the k_off_ jumps when the hydrolysis occurs, which would give spatially varying dwell times and nonexponential distributions as the hydrolysis can occur at random (Poisson process) during the observation time. So it still seems that the data could be consistent with a simple GTP-GDP model, with no need for an additional third state of GDP-Pi.

We agree. We do not intend to conclude that our data are evidence for having detected a GDP-Pi state. We can however also not exclude that a GDP-Pi state contributes to the observed affinity gradient. We believe that the longer version of our revised manuscript helps to prevent this misunderstanding.

b) Figure 4J is unconvincing as the simpler model of only two nucleotide states (GTP and GDP) has not been ruled out. To rule this out it will be necessary to do model-convolution to make it convincing that the analysis method is not yielding spurious results. Even then, the EB binding could be dependent on the local neighborhood of nucleotide state (see Seetapun et al., 2012, Figure 2—figure supplement 1 for conformational spread modeling), which seems reasonable since the binding site is at the interface between tubulins. Overall, model-convolution on the microtubule addition-loss-hydrolysis and EB binding-unbinding to assess the model is needed to rule out the simpler GTP-GDP model. Even then the argument for a GDP-Pi state is not compelling.

We agree also here. Our schematic figure might have been misleading. We have relabeled the scheme, indicating that the colour gradient is indicative of an EB affinity gradient and not intended to make a statement about a GDP-Pi state.

2) The values for KDs of 8 nM and 40 nM for EB3 binding to GTP- and GDP-tubulin, respectively, seem very strong compared to Seetapun et al. values of 3.8 µM and 55 µM, respectively, in vivo. Why is this, and are the in vitro results informative of the tip tracking in living cells?

The Kd values we have measured for EB3 binding are in the usual range of Kd values measured in in vitro experiments, here at the lower end of the spectrum, because we used a lower ionic strength buffer compared to previous experiments. Why the Kd values for binding the growing microtubule end and the GDP lattice as inferred from fluorescence intensity measurements in cells at a single cellular EB1 concentration are different is currently unclear. Since the in vitro situation is simpler, the EB occupancy on the microtubule can be measured at different EB concentrations, resulting in a binding curve that gives confidence in the obtained Kd value. Independent of the discrepancy between the absolute values of the Kd measurements in vitro and in cells, all measurements agree in that the affinity for the growing microtubule end and – as for the first time measured here – for the GTP lattice is higher than for the GDP lattice. In other words, relative comparisons within one system lead to the similar conclusions.

3) How big is the cap with wildtype tubulin in vitro? Note: "cap size" is in the title, but it is not estimated, despite a lot of nice quantitation. Previous estimates put it at ~40 in vitro (at 5 µM wildtype tubulin) and ~750 in vivo (Schek et al., 2006; Seetapun et al., 2012), a disparity that is largely explained by the relative disparity in the net growth rates under these two conditions. These papers should be cited, as previous estimates of cap size. (Note: need to account for tubulin concentration, e.g. 19 µM wildtype tubulin in Figure 3J vs. 5 µM in Schek et al. and 7 µM in. Seetapun et al.).

The cap size was stated in our Figure 3J in units of length for wildtype and E254D microtubules. The spatial decay rate constants are 1/220 nm and 1/690 nm, respectively. The main point made here was that the cap is elongated when GTP hydrolysis is impaired by the E254D mutation. As suggested by the reviewer, the cap lengths for wildtype and E254D microtubules can also be expressed in numbers of tubulin forming the cap (assuming 13pf microtubules, 8 nm tubulin length and a mono-exponential distribution of cap tubulins): 358 and 1120 tubulins, respectively.

Comparing the cap size measured in vitro for wildtype microtubules with the size of 750 tubulins forming the cap of microtubules in growing in cells, one finds that the cap in cells was 2x larger than it is here in vitro for microtubules growing about 3x faster in cells (156 nm/s vs. 57nm/s). Given that cap length can vary easily by a factor of 2 depending on buffer conditions and given that not the same methods for the determination of the decay rate constants were used in our and the Seetapun study, we can conclude that there is good agreement between cap lengths measured in vitro and in cells, as we have already concluded previously citing the Seetapun et al. paper (Maurer et al., 2014).

In Schek et al., comet lengths were not determined. Therefore, a similar comparison is not possible. Schek et al. however reported transient shrinkage episodes during growth corresponding to up to 40 tubulins per shrinkage events which suggested the necessity for long GTP caps as they are observed using EBs as cap markers. We acknowledged this previously by citing the Schek et al. paper (Duellberg et al., 2016). We have included these citations also in the current manuscript.